# PTK2B promotes TBK1 and STING oligomerization and enhances the STING-TBK1 signaling

Yongfang Lin[1,2,3,4,5,6], Jing Yang[1,3,4,5,6], Qili Yang[1,3,4,5], Sha Zeng[2],
Jiayu Zhang [1,3,4,5], Yuanxiang Zhu [2], Yuxin Tong[2], Lin Li [1,3,4], Weiqi Tan[1,3,4],
Dahua Chen[2] ✉ & Qinmiao Sun [1,3,4,5] ✉

TANK-binding kinase 1 (TBK1) is a key kinase in regulating antiviral innate immune responses. While the oligomerization of TBK1 is critical for its full activation, the molecular mechanism of how TBK1 forms oligomers remains unclear. Here, we show that protein tyrosine kinase 2 beta (PTK2B) acts as a TBK1-interacting protein and regulates TBK1 oligomerization. Functional assays reveal that *PTK2B* depletion reduces antiviral signaling in mouse embryonic fibroblasts, macrophages and dendritic cells, and genetic experiments show that *Ptk2b*-deficient mice are more susceptible to viral infection than control mice. Mechanistically, we demonstrate that PTK2B directly phosphorylates residue Tyr591 of TBK1, which increases TBK1 oligomerization and activation. In addition, we find that PTK2B also interacts with the stimulator of interferon genes (STING) and can promote its oligomerization in a kinase-independent manner. Collectively, PTK2B enhances the oligomerization of TBK1 and STING via different mechanisms, subsequently regulating STING-TBK1 activation to ensure efficient antiviral innate immune responses.

Upon viral infection, the host employs pattern recognition receptors (PRRs) to detect viral pathogens and trigger innate immune responses to produce type I interferons (IFNs) and proinflammatory cytokines, subsequently eliminating viruses[1]. Cytosolic PRRs were found to play important roles in recognizing viral nucleic acids. Cytosolic sensors, including retinoic-acid-inducible protein 1 (RIG-I) and melanoma-differentiation-associated gene 5 (MDA5), detect RNA viruses[2], and then recruit adapter protein mitochondrial antiviral signaling protein (MAVS)[3,4] and active TANK-binding kinase 1 (TBK1)/inducible IκB kinase (IKKi) and the IκB kinase (IKK) complex, to induce phosphorylation of the transcription factors interferon regulatory factor 3/7 (IRF3/7) and NF-κB, respectively, leading to production of IFNs and proinflammatory cytokines[1]. For sensing DNA viruses, several cytosolic

receptors, such as DNA-dependent activator of interferon-regulatory factors (DAI)[5], interferon-gamma inducible protein 16 (IFI16)[6], DDX41[7], and cyclic GMP-AMP synthase (cGAS)[8], have been identified. These DNA sensors then recruit a stimulator of IFN genes (STING)[9], subsequently activating TBK1/IKKi and the IKK complex to induce antiviral responses[10,11].

TBK1 is one critical kinase in the innate immune responses[12]. It can phosphorylate IRF3/7, inducing their dimerization and nuclear translocation, which in turn induces IFN production[11,12]. Activation of TBK1 is required for its function. Previous studies revealed that TBK1 phosphorylation is important for its activation[13–15]. For example, AKT1 was found to phosphorylate TBK1 at residue S510 to reduce TBK1 activation[13], and protein phosphatases PPM1A and PPM1B remove TBK

[1]State Key Laboratory of Membrane Biology, Institute of Zoology, Chinese Academy of Sciences, Jia #3 Datun Road, Chaoyang District, 100101 Beijing, China. [2]Institute of Biomedical Research, Yunnan University, 650500 Kunming, China. [3]Institute for Stem Cells and Regeneration, Chinese Academy of Sciences, 100101 Beijing, China. [4]Beijing Institute for Stem Cell and Regenerative Medicine, 100101 Beijing, China. [5]School of Life Sciences, University of Chinese Academy of Sciences, 100049 Beijing, China. [6]These authors contributed equally: Yongfang Lin, Jing Yang. ✉e-mail: chendh@ynu.edu.cn; qinmiaosun@ioz.ac.cn

phosphorylation on Ser residues to reduce its activation[14,15]. In addition to serine residues, TBK1 phosphorylation on tyrosine (Tyr) was also found to regulate its activation. Tyrosine kinases Lck/Hck/Fgr directly phosphorylate TBK1 at residues Y354 and Y394, reducing its activation[16]. In contrast, tyrosine kinase SRC was shown to augment TBK1 autophosphorylation and activation[17]. However, because currently there is no evidence showing that SRC directly phosphorylates TBK1, we speculate that an uncharacterized tyrosine kinase(s) might be involved in directly phosphorylating TBK1 on Tyr residues for its activation, thus positively regulating the antiviral signaling.

Previous studies have shown that protein oligomerization plays a critical role in innate immune responses, for example, RIG-I[18], MDA5[18], cGAS[19], STING[20] and TBK1[21,22] were all found to form oligomers. Protein phosphorylation was shown to alter its oligomeric status and potentially influence its function in the cells. TBK1 phosphorylates STING and promotes its oligomerization[14,23], whereas PPM1A reduces STING oligomerization[14]. GSK3β was reported to promote TBK1 oligomerization, subsequently affecting autophosphorylation and activation[22], whereas tyrosine kinase Lck/Hck/Fgr phosphorylates TBK1 on Tyr residues, reducing TBK1 dimerization and activation[16]. These findings suggest that phosphorylation of TBK1 probably alters its structural conformation, and then regulates its oligomer formation, modulating its activity. However, the precise regulatory relationships between TBK1 phosphorylation and oligomerization are still largely unknown.

Protein tyrosine kinase 2 beta (PTK2B) is a member of the focal adhesion kinase family, which is composed of PTK2B and PTK2 and plays important roles in a variety of biological processes. Previous studies have shown that PTK2B regulates cancer development by mediating cell morphology, migration, proliferation, adhesion and survival[24]. PTK2B is also involved in regulating T cell activation[25]. In addition, PTK2B is highly expressed in immune cells and can be activated by antigen receptor, or chemokine receptor stimulation[25]. PTK2B has been demonstrated to regulate inflammasome responses by augmenting proinflammatory gene expression induced by tumor necrosis factor-alpha (TNF-α) and interleukin (IL)−1β[26] or phosphorylating apoptosis-associated speck-like protein containing a CARD (ASC) to modulate ASC speck formation and inflammasome activation[27]. However, despite the aforementioned role of PTK2B in regulating inflammasome responses, whether PTK2B mediates antiviral signaling remains largely unknown.

In this study, we found that PTK2B interacted with TBK1 and STING, and regulated the activation of TBK1. Ptk2b-knockout mice were more susceptible to lethal herpes simplex virus type 1 (HSV-1) and vesicular stomatitis (VSV) infection than control mice. Furthermore, we found that PTK2B directly phosphorylated TBK1 at residue Y591, augmenting its oligomer formation and activation. In addition, PTK2B could promote STING oligomerization in a kinase-independent manner. Together, PTK2B plays an important role in modulating antiviral signaling by regulating TBK1 and STING oligomerization and activation.

## Results

### PTK2B interacts with TBK1 and STING

TBK1 plays a critical role in regulating innate immune responses. To search for regulatory factors of TBK1 activation, we screened for proteins that interact with TBK1 using co-immunoprecipitation (Co-IP) experiments combined with mass spectrometry analysis. Due to the difficulty of obtaining mouse macrophages RAW 264.7 cells stably expressing TBK1, HEK293T cells expressing S protein-Flag-Streptavidin binding peptide (SFB)-tagged mouse TBK1 or control vector cells were employed. TBK1 was purified using S-protein agarose beads, and then the protein was incubated with cell lysates from RAW 264.7 cells, followed by Co-IP assays and mass spectrometry analysis. We identified a number of interesting candidate proteins that interacted with TBK1 (Supplementary Table S1). Of note, PRMT1 and HNRNPA2B1 from the

list have been reported to interact with TBK1[28,29]. We were particularly interested in one candidate, PTK2B, because it is a tyrosine kinase and phosphorylation of TBK1 is important for its activity.

To confirm the association between PTK2B and TBK1, we overexpressed PTK2B and TBK1 in HEK293T cells and conducted Co-IP assays. As shown in Fig. 1a, overexpressed Myc-tagged PTK2B was associated with Flag-tagged TBK1. Next, we examined whether PTK2B specifically interacted with TBK1. We overexpressed Myc-tagged PTK2B with Flag-tagged TBK1 or other important components of the cGAS-STING pathway, including cGAS, STING and IRF3. Co-IP results showed that PTK2B also interacted with STING and IRF3, but not with cGAS (Fig. 1b). Notably, the interaction between PTK2B and IRF3 was much weaker than those with STING and TBK1. To further examine the interactions of PTK2B with STING, TBK1 and IRF3, we next performed endogenous Co-IP experiments with PTK2B antibody in RAW 264.7 cells with or without HSV1-GFP infection and found that PTK2B only pulled down STING and TBK1 regardless of the HSV1-GFP infection state (Fig. 1c). Of note, Co-IP results showed that HSV1-GFP infection markedly enhanced the association of TBK1 and PTK2B, but it had no apparent effect on enhancing the STING-PTK2B interaction. We reasoned that this is probably due to the decreased expression of STING upon HSV1-GFP infection. Previous studies have suggested the degraded STING protein induced by virus infection is blocked by MG132 (a proteasome inhibitor)[30]. To test how virus infection affects the STING-PTK2B interaction, we pretreated the cells with MG132 and then infected with HSV1-GFP. We found that the MG132 treatment also failed to increase the association of PTK2B with STING upon virus infection (Supplementary Fig. S1a), suggesting that virus infection has no apparent effect on the PTK2B-STING association. Consistent with Co-IP results, immunostaining assays showed that co-localization of TBK1 and PTK2B were significantly enhanced in SV40-immortalized mouse embryonic fibroblasts (MEFs) upon HSV-1 infection. Interestingly, we noted that both TBK1 and PTK2B formed granules upon HSV-1 infection (Fig. 1d). Moreover, by performing in vitro Co-IP assays using purified glutathione S-transferase-tagged (GST)-tagged PTK2B and His-tagged TBK1 (residues 1–657) expressed in Escherichia coli (E. coli), we found that His-TBK1, but not His-GFP, efficiently pulled down GST-PTK2B (Fig. 1e). These findings suggest that PTK2B directly binds to TBK1. Additionally, direct interaction between PTK2B and STING in vitro was also detected when we employed purified GST-PTK2B and His-STING (residues 153–379) expressed in E. coli (Fig. 1f). These results suggest that PTK2B directly interacts with TBK1 and STING, and potently regulates STING-TBK1 signaling. In addition, Co-IP results indicated that TBK1, but not RIG-I and MAVS, was pulled down by PTK2B when cells were infected with or without vesicular stomatitis mutant virus (an RNA virus), which carries a single amino acid deletion (Met51) in the matrix protein of VSV-GFP (VSVΔM51-GFP)[31] (Supplementary Fig. S1b). These results suggest that PTK2B associates with TBK1 and potently regulates antiviral signaling against RNA virus infection.

Next, we characterized the association of PTK2B with TBK1 or STING by conducting domain-mapping experiments. TBK1 has three domains: an N-terminal kinase domain (KD), a central ubiquitin-like domain (ULD) and C-terminal two coiled-coil domains (CCD)[32]. We generated truncated variants of TBK1: TBK1-KD (residues 1–307), TBK1-ULD (residues 308–407) and TBK1-CCD (residues 408–729). As shown in Fig. 1g, PTK2B is associated with the KD and ULD of TBK1, but not the CCD. PTK2B has N- and C-terminal domains and a central kinase domain (KD)[33]. Co-IP results indicated that the C-terminal domain of PTK2B was not required for the association between PTK2B and TBK1 (Fig. 1h). Additionally, we generated two truncated variants of STING: STING-N (residues 1–136) and STING-C (residues 137–379)[34] and found that both the N- and C-terminal domains of STING associated with PTK2B (Supplementary Fig. S1c), and both the N-terminal domain and the KD of PTK2B, but not its C-terminal domain, interacted with STING

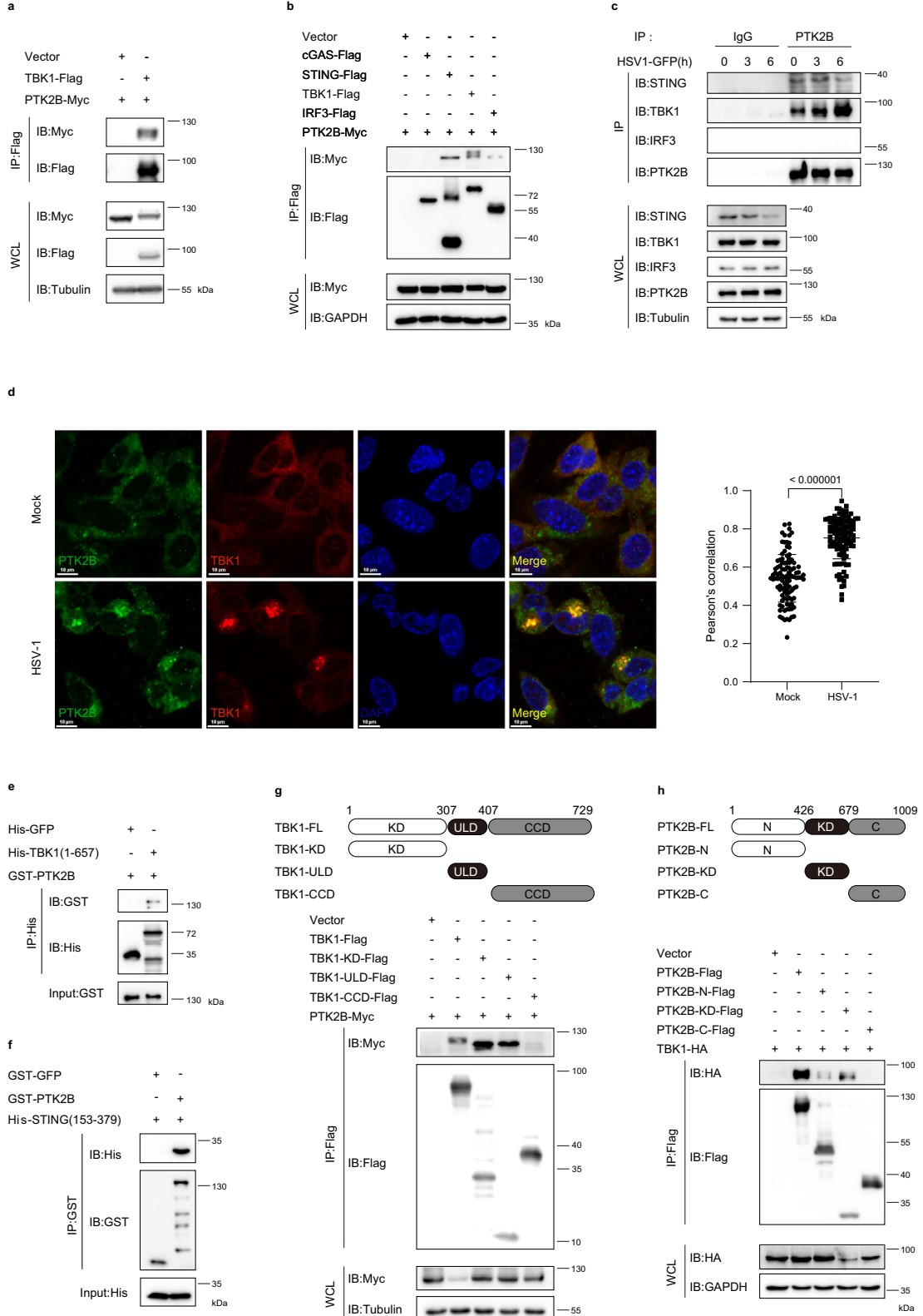

(Supplementary Fig. S1d). These data suggest that PTK2B interacts with TBK1 and STING in a domain-dependent manner.

**PTK2B depletion attenuates the activation of antiviral signaling**
Considering that PTK2B is associated with TBK1 and STING, next we examined whether PTK2B regulates STING-TBK1 signaling. First, we knocked down PTK2B in human macrophage THP1 cells by using lentivirus-mediated short hairpin RNA (shRNA). Quantitative PCR (qPCR) analysis showed that PTK2B knockdown significantly attenuated the transcriptional levels of antiviral genes such as *IFNB1*, *IFIT1* and *CXCL10* induced by HSV1-GFP infection (Fig. 2a–d) or by transfection with human telomeric DNA (HT-DNA) (Supplementary Fig. S2a–c). To determine whether PTK2B specifically mediates the innate immune response to DNA viruses, we infected PTK2B-

**Fig. 1 | PTK2B interacts with TBK1 and STING. a** HEK293T cells were transfected with Myc-tagged PTK2B and Flag-tagged TBK1 or empty vector and lysed 24 h after transfection for co-immunoprecipitation (Co-IP) with anti-Flag M2 beads, and then the pulled-down proteins were analyzed by immunoblotting. **b** HEK293T cells were transfected with the indicated expression plasmids. Co-IP assays were performed with anti-Flag M2 beads and analyzed by immunoblotting. **c** RAW 264.7 cells were mock infected or infected with HSV1-GFP for the indicated times. The cell lysates were immunoprecipitated with anti-PTK2B antibody or control IgG and analyzed by immunoblotting. **d** Immortalized mouse embryonic fibroblasts (MEFs) were mock infected or infected with HSV-1 for 6 h. Cells were then fixed, stained with PTK2B (green) and TBK1 (red) antibodies, then imaged by confocal microscopy (left). Scale bars, 10 μm. Relative co-localization of PTK2B and TBK1 was quantified with Pearson's correlation coefficient by using NIKON NIS-Elements Analysis software (right), the cells from mock infected (n = 104) or infected (n = 110) group were analyzed. **e** A mixture of purified GST-PTK2B and His-TBK1 (residues 1–657) or His-GFP,

expressed in *E. coli*, was pulled down with Ni-Sepharose beads, and then analyzed by immunoblotting. **f** A mixture of purified His-STING (residues 153–379) and GST-PTK2B or GST-GFP, expressed in *E. coli*, was pulled down with Glutathione-Sepharose beads, and then analyzed by immunoblotting. **g** Schematic diagram of TBK1 domains (top). HEK293T cells were co-transfected with PTK2B-Myc and TBK1-Flag or its truncated mutants as indicated. Co-IP assays were performed with anti-Flag M2 beads and the pulled-down proteins were analyzed by immunoblotting (bottom). **h** Schematic diagram of PTK2B domains (top). HEK293T cells were co-transfected with TBK1-HA and PTK2B-Flag or its truncated mutants as indicated. Co-IP assays were performed with anti-Flag M2 beads and the pulled-down proteins were analyzed by immunoblotting (bottom). Data shown in (**a**–**c**, **e**–**h**) are from one representative of two independent experiments with similar results. Data shown in (**d**) are from one representative experiment of three independent experiments (mean ± SD), two-tailed Student's *t*-test. Source data are provided as a Source Data file.

knockdown THP1 cells or control cells with VSVΔM51-GFP and obtained similar results to those following HSV1-GFP infection (Supplementary Fig. S2d, e). Consistently, we observed that PTK2B knockdown reduced the mRNA level of *IFNB1* induced by poly(I:C), a synthetic dsRNA that mimics RNA virus infection (Supplementary Fig. S2f). Next, we conducted immunoblotting assays and found that PTK2B knockdown reduced the levels of phosphorylated STING, TBK1 and IRF3 induced by infection with HSV1-GFP (Fig. 2e). Consistently, the levels of phosphorylated TBK1 and IRF3 induced by VSVΔM51-GFP infection were decreased in PTK2B-knockdown THP1 cells compared with those in the control cells (Supplementary Fig. S2g). In addition, we employed antisense oligonucleotides (ASOs) to reduce PTK2B expression. As shown in Supplementary Fig. S2h, i, PTK2B ASOs treatment downregulated levels of PTK2B protein, and reduced the levels of *IFNB1* mRNA induced by the infection with HSV1-GFP or VSV-GFP. These findings indicate that PTK2B plays a positive role in regulating innate immune responses to viral infection.

Next, we investigated whether PTK2B plays a role in regulating antiviral signaling in mouse cells. First, we overexpressed PTK2B in MEFs. qPCR results indicated that PTK2B overexpression significantly enhanced the mRNA levels of *Ifnb1*, *Ifit1* and *Cxcl10* induced by infection with HSV1-GFP or VSVΔM51-GFP (Fig. 2f–h, Supplementary Fig. S3a–d) in MEFs. Consistently, PTK2B overexpression not only augmented the levels of phosphorylated STING, TBK1 and IRF3 induced by HSV1-GFP infection but also increased the levels of phosphorylated TBK1 and IRF3 induced by VSVΔM51-GFP infection (Supplementary Fig. S3e, f). Second, we knocked down Ptk2b in mouse macrophages RAW 264.7 cells and found that Ptk2b knockdown significantly impaired mRNA levels of *Ifnb1*, *Ifit1* and *Cxcl10* induced by infection with HSV1-GFP (Supplementary Fig. S3g–j). Similarly, Ptk2b knockdown significantly attenuated the mRNA level of *Ifnb1* stimulated by VSVΔM51-GFP (Supplementary Fig. S3k). Taken together, these findings suggest that PTK2B functions as a conserved regulator of antiviral signaling.

To further validate the biological role of PTK2B in antiviral signaling, we generated Ptk2b-knockout RAW 264.7 cells by employing the CRISPR−Cas9 system[35]. qPCR results showed that Ptk2b deficiency dramatically reduced mRNA levels of *Ifnb1*, *Ifit1* and *Cxcl10* stimulated by HSV1-GFP or VSVΔM51-GFP infection (Fig. 2i–k, Supplementary Fig. S3l–n). Consistently, we found that Ptk2b knockout decreased the levels of phosphorylated TBK1 and IRF3 induced by HSV1-GFP or VSVΔM51-GFP infection (Fig. 2l, Supplementary Fig. S3o).

Given that PTK2B plays a positive role in antiviral signaling, we next tested whether PTK2B modulates viral replication. As shown in Fig. 2m, the titers of HSV1-GFP or VSVΔM51-GFP were significantly higher in Ptk2b-knockout RAW 264.7 cells than in wild-type control cells. Collectively, these data indicate that PTK2B is vital for effective activation of antiviral signaling.

## PTK2B is critical for antiviral signaling in vivo

To further explore the role of PTK2B in host defense against viral infection, we isolated primary MEFs from Ptk2b-deficient and littermate control embryos by crossing Ptk2b heterozygous mice. qPCR assays showed that Ptk2b deficiency significantly attenuated the mRNA levels of *Ifnb1*, *Ifit1* and *Cxcl10* stimulated by HSV1-GFP and VSVΔM51-GFP infection (Fig. 3a–c, Supplementary Fig. S4a–c). Consistently, immunoblotting results showed that the levels of phosphorylated STING, TBK1 and IRF3 stimulated by HSV1-GFP infection were decreased in Ptk2b-deficient cells compared with wild-type control cells (Fig. 3d). To determine the specific role of PTK2B in antiviral signaling and whether its kinase activity is required for its function, we performed rescue experiments in Ptk2b-deficient cells and found that, while ectopic expression of wild-type PTK2B almost completely restored the decreased expression of *Ifnb1* induced by HSV1-GFP, its kinase-inactive mutant PTK2B K457R[36], in which lysine (K) residue 457 was mutated to arginine (R), only partially rescued the function of PTK2B (Fig. 3e). This finding indicated that the kinase activity of PTK2B is required for its function in regulating antiviral signaling. Next, we examined the effect of PTK2B in regulating antiviral signaling in bone marrow-derived macrophages (BMDMs) isolated from Ptk2b-deficient and control mice. We observed that Ptk2b knockout significantly reduced the mRNA levels of antiviral genes stimulated by HSV1-GFP or VSVΔM51-GFP (Fig. 3f–h, Supplementary Fig. S4d, e). Consistently, Ptk2b deficiency reduced the levels of phosphorylated STING, TBK1 and IRF3 stimulated by HSV1-GFP infection (Supplementary Fig. S4f). In addition, we found that Ptk2b deficiency significantly reduced mRNA levels of *Ifnb1*, *Ifit1* and *Cxcl10* induced by HSV1-GFP or VSVΔM51-GFP in bone marrow-derived dendritic cells (BMDC) (Supplementary Fig. S4g–l). These findings demonstrate that PTK2B plays an important role in regulating antiviral signaling in mouse primary MEFs, BMDMs and BMDCs.

To investigate the role of PTK2B in immune defense against viral infection in vivo, we intravenously infected Ptk2b-deficient and control mice with HSV-1. Enzyme-linked immunosorbent assays (ELISAs) indicated that Ptk2b deficiency significantly reduced the levels of serum IFNβ, IL-6, TNFα (Fig. 3i, Supplementary Fig. S5a, b). In agreement with this, more HSV-1 viral particles were produced in Ptk2b−/− mice than in the wild-type mice (Fig. 3j, Supplementary Fig. S5c), and Ptk2b knockout significantly reduced the survival rates of mice after HSV-1 infection (Fig. 3k). Consistently, greater infiltration of immune cells (e.g., Dendritic cells and Neutrophils cells) and tissue damage were observed in the lungs of infected Ptk2b−/− mice than in wild-type mice (Supplementary Fig. S5d–f and Fig. 3l). Additionally, we found that Ptk2b deficiency significantly decreased the production of IFNβ and IL-6 and the survival rates of mice upon VSV virus infection (Supplementary Fig. S5g–i). Collectively, these results suggest that PTK2B plays a critical role in immune defense against viral infection in vivo.

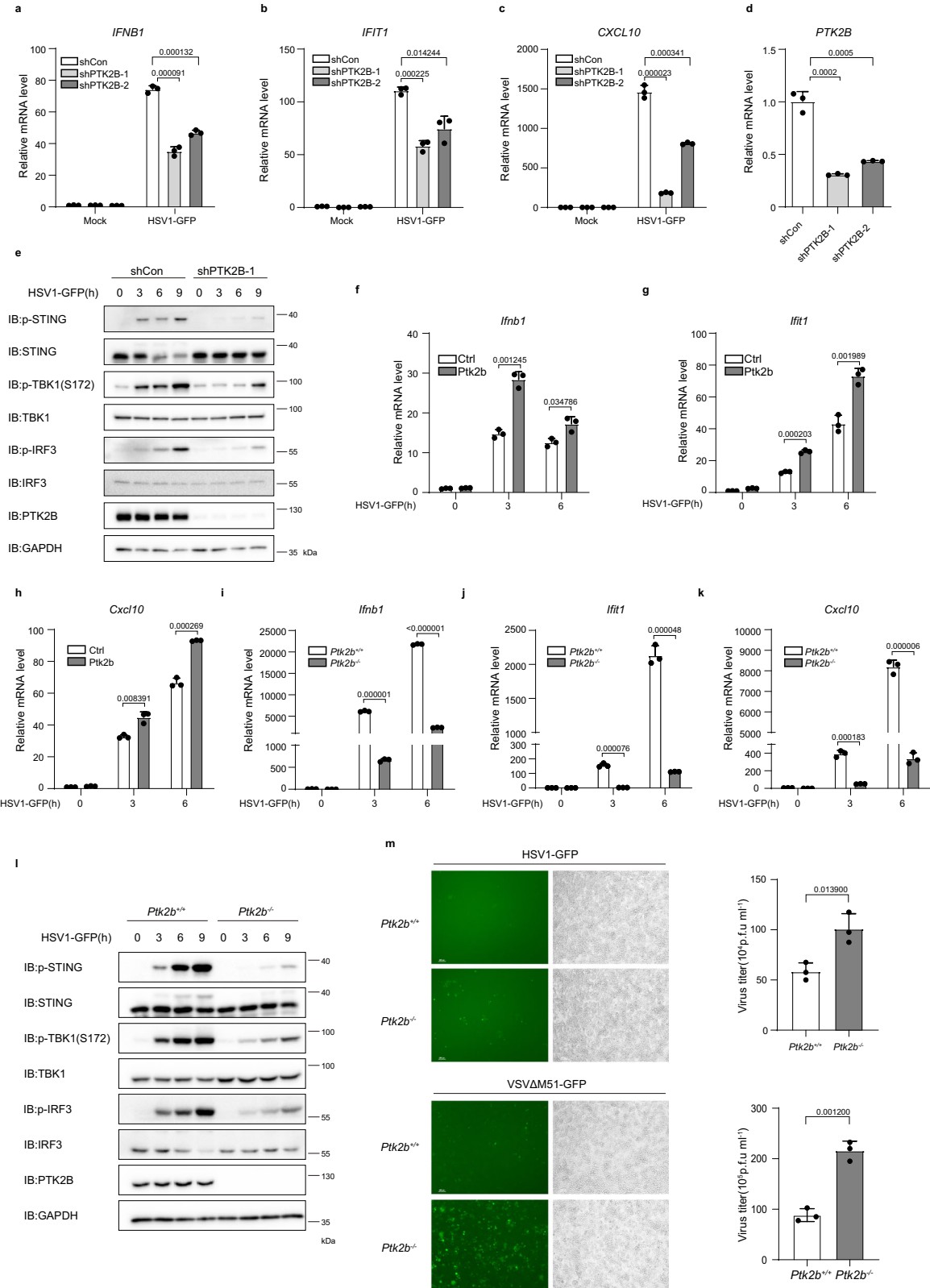

## PTK2B enhances TBK1 activation by regulating tyrosine phosphorylation

The experiments described above indicated that PTK2B plays an important role in regulating innate immune responses to virus infection, and its kinase activity is required for its full function. Next, we explored how PTK2B mediates antiviral signaling. Given that PTK2B interacts with TBK1, we reasoned that PTK2B mediated antiviral

signaling largely through regulating TBK1 activation. To test this idea, we investigated whether and how PTK2B modulates TBK1 activity. Immunoblotting assays indicated that ectopic wild-type PTK2B, but not its kinase-inactive mutant PTK2B-K457R augmented the level of S172-phosphorylated TBK1, which indicates TBK1 activity[37], in HEK293T cells (Fig. 4a). These data suggested that PTK2B increased the activation of TBK1 in a kinase-dependent manner. Next, we

**Fig. 2 | PTK2B plays a positive role in regulating antiviral signaling. a−d** THP-1 cells were infected with short hairpin RNA (shRNA) lentivirus targeting two different regions of PTK2B (shPTK2B-1, shPTK2B-2) or negative control (shCon), followed by infection with HSV1-GFP for 6 h. Quantitative PCR (qPCR) assays were performed to measure the mRNA levels of *IFNB1* (**a**), *IFIT1* (**b**), *CXCL10* (**c**) and *PTK2B* (**d**). **e** THP-1 cells stably expressing shRNA targeting PTK2B or control cells were infected with HSV1-GFP for the indicated times, followed by immunoblotting. **f−h** Immortalized mouse embryonic fibroblasts (MEFs) stably expressing PTK2B or control cells were infected with HSV1-GFP for 3 and 6 h. *Ifnb1* (**f**), *Ifit1* (**g**) and *Cxcl10* (**h**) mRNA levels were measured by qPCR. **i−k** *Ptk2b*[+/+] and *Ptk2b*[−/−] RAW 264.7 cells were infected with HSV1-GFP for 3 and 6 h and then analyzed by qPCR to quantify *Ifnb1* (**i**), *Ifit1* (**j**) and *Cxcl10* (**k**) mRNA levels. **l** *Ptk2b*[+/+] and *Ptk2b*[−/−] RAW 264.7 cells were infected with HSV1-GFP for the indicated times and then analyzed by immunoblotting with the indicated antibodies. **m** *Ptk2b*[+/+] and *Ptk2b*[−/−] RAW 264.7 cells were infected with HSV1-GFP for 24 h, or VSVΔM51-GFP for 12 h. The cells were imaged by fluorescence microscopy (left, Scale bars, 200 μm.), or the culture supernatants were harvested to quantify the viral titer using a plaque assay (right). Data shown in (**a−d**, **f−k**, **m**) are from one representative experiment of three independent experiments (mean ± SD, *n* = 3 independent samples), two-tailed Student's *t*-test. Data shown in (**e**, **l**) are one representative of two independent experiments with similar results. Source data are provided as a Source Data file.

examined whether PTK2B regulated Tyr phosphorylation of TBK1. First, we co-transfected Flag-tagged TBK1 with Myc-tagged PTK2B or its kinase-inactive mutant PTK2B-K457R into HEK293T cells and then pulled down with anti-Flag M2 beads. Immunoblotting assays with anti-phospho-Tyr antibody showed that PTK2B dramatically increased Tyr phosphorylation of TBK1, whereas its kinase-inactive mutant had no effect (Fig. 4b). Second, we performed in vitro Tyr phosphorylation of TBK1 assay using purified TBK1, and PTK2B-KD or PTK2B-KD (K457R), expressed in *E. coli*. As shown in Fig. 4c, wild-type PTK2B, but not its kinase-inactive mutant, directly phosphorylated TBK1 in vitro. Next, we examined the effect of PTK2B in regulating TBK1 activation during viral infection. We first tested whether HSV1-GFP infection induced the activation of PTK2B, which was detected by Y402-phosphorylated PTK2B in RAW 264.7 cells[38]. Immunoblotting results demonstrated that similar to the activation of TBK1 and IRF3, the activation of PTK2B was enhanced upon HSV1-GFP infection in a time-dependent manner (Fig. 4d). Then, we examined whether Tyr phosphorylation of TBK1 occurred upon HSV1-GFP infection. As shown in Fig. 4e, HSV1-GFP infection induced robust Tyr phosphorylation of TBK1 in wild-type RAW 264.7 cells, whereas Ptk2b deficiency reduced this effect. These data suggest that HSV1-GFP infection induced PTK2B activation and subsequently increased Tyr phosphorylation of TBK1. Combined with our finding that PTK2B deficiency attenuated the levels of phosphorylated TBK1 induced by HSV1-GFP (Fig. 2l), these results demonstrate that PTK2B phosphorylates TBK1 at Tyr residue (s), subsequently enhancing TBK1 activation.

To investigate how PTK2B phosphorylates TBK1, we searched for Tyr residues of TBK1 that are phosphorylated by PTK2B. Flag-tagged TBK1 was co-transfected into HEK293T cells with Myc-tagged PTK2B or control vector; cell lysates were immunoprecipitated with anti-Flag M2 beads, followed by mass spectrometric analysis. The results from mass spectrometry assays showed that four Tyr residues at Y394, Y577, Y591 and Y592 were phosphorylated by PTK2B (Supplementary Fig. S6a). To confirm these phosphorylation sites, we generated four point mutants of TBK1, Y394F, Y577F, Y591F and Y592F, in which the Tyr residues at positions 394, 577, 591 and 592 were replaced by phenylalanine (F), respectively. In vivo kinase assays in HEK293T revealed that Y591F, but not the other TBK1 mutants, lost enhanced Tyr phosphorylation of TBK1 by PTK2B (Fig. 4f). These data indicate that PTK2B phosphorylates TBK1 predominantly at Y591. To validate this result, we generated an antibody specific for TBK1 phosphorylation at Y591 (pTBK1-Y591) and found that PTK2B enhanced the phosphorylation level of wild-type TBK1 at residue Y591, but much weaker for its Y591F mutant (Fig. 4g). In addition, we observed that the level of phosphorylation at Y591 induced by HSV1-GFP infection was much higher in wild-type RAW 264.7 cells than in Ptk2b-deficient cells (Fig. 4h). Next, we examined the role of Y591 phosphorylation by PTK2B on the activation of TBK1. Immunoblotting results showed that the increased activation of TBK1-Y591F mutant by PTK2B was much lower than that for the wild-type and the other point mutants (Fig. 4i). Consistent results were obtained when PTK2B and wild-type TBK1 or its mutants were co-expressed in TBK1 knockout MEF cells, followed by HSV1-GFP infection (Supplementary Fig. S7a). Notably, we found that PTK2B expression

was markedly reduced in TBK1-deficient MEF cells, compared with control cells (Supplementary Fig. S7b). In addition, we generated two TBK1 mutants Y591E and Y591D by mutating residue Y591 to glutamine (E) or aspartic acid (D), respectively, to mimic constitutive phosphorylation of Y591. As shown in Fig. 4j, expression of PTK2B, at least in part, increased the activation of the TBK1-Y591E mutant, and it had no apparently increased effect on the activation of TBK1-Y591D. We also noted that TBK1-Y591E or TBK1-Y591D alone did not have the increased activity, suggesting that such phosphorylation mimics of Y591 were not sufficient to increase TBK1 activation. Moreover, we observed that wild-type TBK1, but not its mutants Y591F and Y591E, increased levels of phosphorylated IRF3 and *Ifnb1* production induced by HSV1-GFP infection when PTK2B was co-expressed in TBK1-deficient MEF cells (Supplementary Fig. S7c, d). Previous studies have reported that the phosphorylated residue of interest is normally mutated to a negatively charged residue (e.g., aspartate or glutamate); however, these amino acids fail to recapitulate the true steric and charge-based nature of the phosphoryl-modification[39,40]. Nevertheless, these findings suggest that Y591 of TBK1 is phosphorylated by PTK2B and this phosphorylation plays an important role in regulating the full activation of TBK1.

## PTK2B enhances TBK1 oligomerization in a kinase-dependent manner

Next, we explored how PTK2B enhances the activation of TBK1. Previous studies have shown that the dimerization and oligomerization of TBK1 are critical for its activation[21,22]. Thus, considering that PTK2B enhanced TBK1 activation, we investigated whether PTK2B modulated TBK1 oligomerization. First, we employed semi-denaturing detergent agarose gel electrophoresis (SDD-AGE) to detect the oligomerization of TBK1 and found that wild-type PTK2B, but not the kinase-inactive mutant, dramatically enhanced TBK1 oligomerization. This enhanced TBK1 oligomerization by PTK2B was lost when the cell lysates were treated with dithiothreitol (DTT), which reduces disulfide bonds (Fig. 5a). Next, we examined whether PTK2B induced TBK1 oligomerization in vitro by employing purified TBK1, PTK2B-KD and PTK2B-KD (K457R) expressed in *E. coli* and found that PTK2B, but not PTK2B-K457R, remarkably increased TBK1 oligomerization (Fig. 5b). These results suggest that PTK2B enhanced TBK1 oligomerization in a kinase-dependent manner. Additionally, we employed Native Polyacrylamide Gel Electrophoresis (Native-PAGE) to detect TBK1 dimers and found that TBK1 dimerization induced by HSV1-GFP was markedly attenuated in *Ptk2b*[−/−] RAW 264.7 cells compared with wild-type control cells (Fig. 5c). Consistently, we observed that PTK2B deficiency reduced the formation of TBK1 granules induced by HSV1-GFP infection (Fig. 5d). Next, we examined whether PTK2B modulates TBK1 oligomerization induced by RNA virus infection. We noted that although it was difficult to detect the dimer of TBK1 when the anti-TBK1 antibody was used in the Native-PAGE assays, the dimer of phosphorylated TBK1 could be reliably detected when the anti-phosphorylated TBK1 antibody was used. We observed that Ptk2b deficiency reduced the dimer of phosphorylated TBK1 induced by the infection of Sendai virus (Supplementary Fig. S8a). Additionally, immunostaining assays displayed that loss of PTK2B decreased the granule formation of TBK1 induced by the

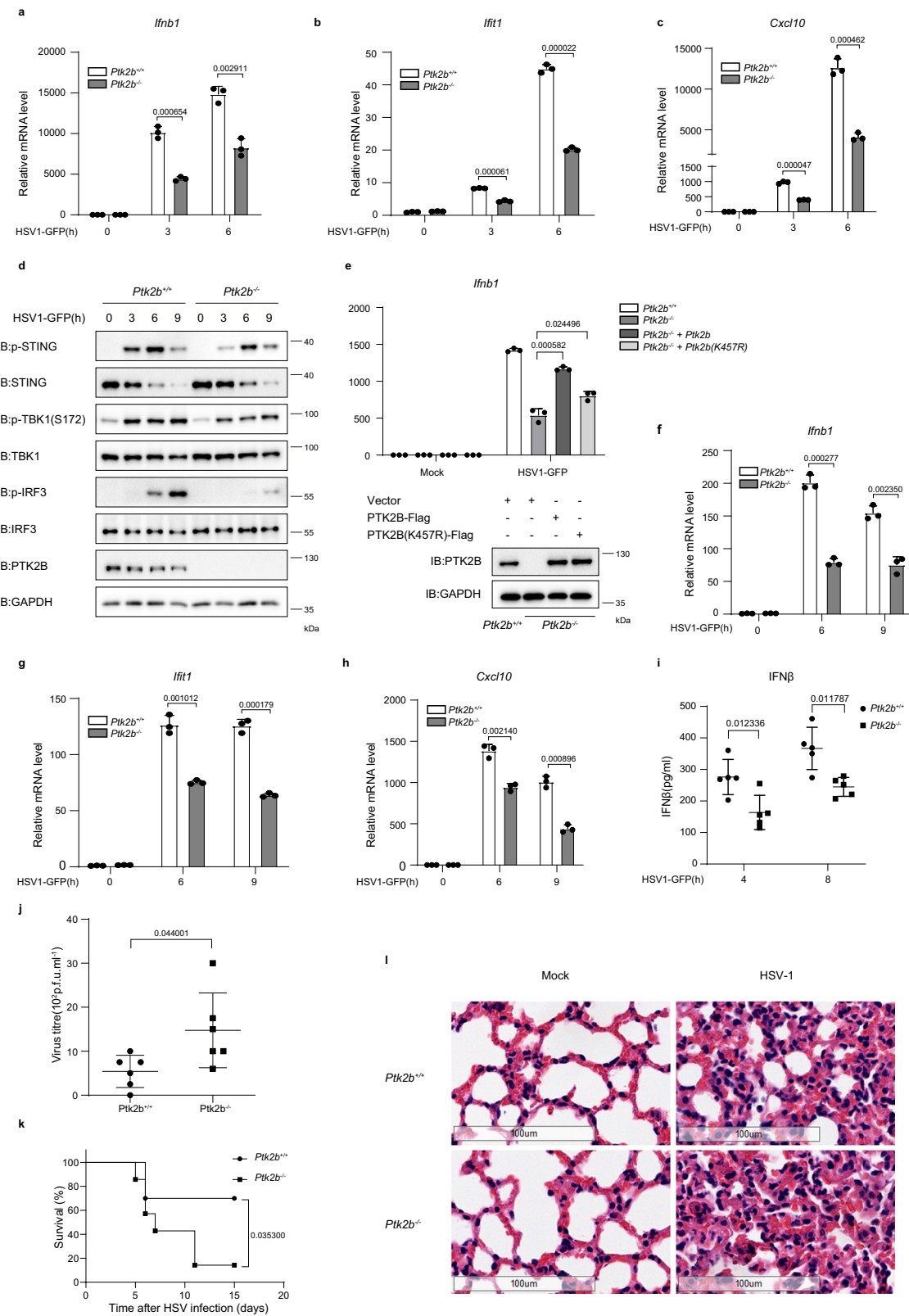

transfection of poly(I:C), which is used to mimic RNA virus infection (Fig. 5e). These findings indicate that PTK2B regulates TBK1 oligomerization induced by virus infection. Collectively, the data suggest that PTK2B phosphorylates TBK1, and then augments TBK1 oligomerization, subsequently increasing TBK1 activity.

Given that PTK2B could augment TBK1 oligomerization in a kinase-dependent manner and that residue Y591 of TBK1 was the primary residue phosphorylated by PTK2B, we next examined the effect of PTK2B on the oligomerization of TBK1-Y591F and other Tyr point mutants of TBK1. As shown in Fig. 5f, unlike wild-type TBK1, Y394F, Y577F and Y592F mutants, the oligomerization of TBK1-T591F induced by PTK2B was strikingly decreased. We also found that PTK2B only partially elevated the oligomerization of TBK1-Y591E (Fig. 5g). Moreover, we performed immunostaining assays to examine the

**Fig. 3 | PTK2B deficiency reduces innate immune responses to virus infection in vivo. a–c** Primary *Ptk2b*[+/+] and *Ptk2b*[−/−] MEFs were infected with HSV1-GFP for the indicated times and then lysed for the quantification of *Ifnb1* (**a**), *Ifit1* (**b**) and *Cxcl10* (**c**) mRNA levels by qPCR. **d** Primary *Ptk2b*[+/+] and *Ptk2b*[−/−] MEFs were infected with HSV1-GFP for the indicated times and then analyzed by immunoblotting. **e** Primary *Ptk2b*[+/+] and *Ptk2b*[−/−] MEFs were infected with lentivirus expressing wild-type PTK2B, PTK2B(K457R) or empty vector, and then infected with HSV1-GFP for 6 h. The cells were harvested for qPCR to measure the mRNA levels of *Ifnb1* (top) or for immunoblotting with the indicated antibodies (bottom). **f–h** *Ptk2b*[+/+] and *Ptk2b*[−/−] bone marrow-derived macrophages (BMDMs) were infected with HSV1-GFP for the indicated times, followed by qPCR to measure the mRNA levels of *Ifnb1* (**f**), *Ifit1* (**g**) and *Cxcl10* (**h**). **i** 12-week-old *Ptk2b*[+/+] and *Ptk2b*[−/−] mice were infected with HSV-1 via tail vein injection at $4 \times 10^7$ pfu per mouse. Sera were collected at 4 and 8 h of infection to measure IFNβ levels by ELISA. *n* = 5 mice for each group. **j** 12-week-old

*Ptk2b*[+/+] and *Ptk2b*[−/−] mice were infected with HSV-1 via tail vein injection at $4 \times 10^7$ pfu per mouse. Sera were collected at 24 h of infection to measure viral titers using a plaque assay. *n* = 6 mice for each group. **k** 12-week-old *Ptk2b*[+/+] (*n* = 10) and *Ptk2b*[−/−] (*n* = 7) mice were infected with HSV-1 via tail vein injection at $4 \times 10^7$ pfu per mouse and the survival of mice was monitored for 15 days. **l** 12-week-old *Ptk2b*[+/+] and *Ptk2b*[−/−] mice were infected with HSV-1 via tail vein injection at $3 \times 10^7$ pfu per mouse for 4 days. Sections of lung from infected mice were analyzed using hematoxylin and eosin staining. Scale bars, 100 μm. Data shown in (**a**–**c**, **e**–**h**) are from one representative experiment of three independent experiments (mean ± SD, *n* = 3 independent samples), two-tailed Student's *t*-test. Data shown in (**d**, **i**, **j**, **l**) are one representative of two independent experiments with similar results. The log-rank (Mantel–Cox) test was used in Data (**k**). Source data are provided as a Source Data file.

granule formation of TBK1 wild-type and TBK1-Y591F mutant upon HSV-1 infection in TBK1-deficient MEFs when cells were co-expressed with PTK2B, we found that TBK1-Y591F mutant formed fewer and smaller granules induced by HSV-1 infection, compared to the TBK1 wild-type (Supplementary Fig. S8b). These results collectively indicate that phosphorylation of residue Tyr591 in TBK1 is critical for the enhanced oligomerization of TBK1 induced by PTK2B.

Having seen that PTK2B kinase activity played an important role in regulating TBK1 oligomerization, we next examined how PTK2B was activated. Given that PTK2B and TBK1 could co-localize together and form cellular granules upon HSV-1 infection (Fig. 1d), we reasoned that PTK2B could potentially undergo oligomerization, in response to virus infection. To test this idea, we performed SDD-AGE assays and observed that HSV-1 infection induced phosphorylated PTK2B oligomerization in a time-dependent manner, which is consistent with the activation of PTK2B (Supplementary Fig. S9a). Moreover, we employed the "ddFP" system to study the behavior of PTK2B and TBK1 in cells. The ddFP system has two parts of monomers: one monomer (A) contains a chromophore that is quenched under the monomeric state, and the other one (B) does not form a chromophore, but it can substantially increase fluorescence of the part A when A and B form a heterodimer. We fused PTK2B with GA or GB to generate two constructs, PTK2B-GA and PTK2B-GB, respectively. As shown in Fig. 5h, in the absence of viral infection, no apparent green signal was detected in cells with co-expression of PTK2B-GA and PTK2B-GB. However, strong green signals showing PTK2B self-interaction were detected from cellular granules, upon HSV-1 infection. In addition, TBK1 oligomerization signals were also detected to be overlapped with PTK2B granules on the Golgi. TBK1 oligomerization is the hallmark of TBK1 activation. Thus, our findings uncover a mechanism by which the PTK2B oligomerization induced by viral infection plays a positive role in regulating antiviral signaling by promoting TBK1 phosphorylation and oligomerization.

### PTK2B can enhance STING oligomerization in a kinase-independent manner

Our Co-IP results described above indicated that PTK2B also associates with STING. We also found that STING can translocate to Golgi from ER and form granules upon HSV-1 infection, which were consistent with the previous findings[41], and that the granules of PTK2B co-localized with the STING signals on Golgi in MEF cells upon HSV-1 infection (Supplementary Fig. S10a, b). Moreover, we observed that upon HSV-1 infection, TBK1, STING and PTK2B co-localized together and formed foci on Golgi in MEFs (Supplementary Fig S10c). Considering that PTK2B enhanced TBK1 oligomerization and that STING oligomerization plays an important role in STING-mediated signaling[11,23], we next investigated whether PTK2B also regulates STING oligomerization. First, we employed SDD-AGE to detect STING oligomerization and found that overexpression of PTK2B dramatically enhanced STING oligomerization, this

influence was abolished when the cell lysates were treated with DTT (Fig. 6a). Consistently, SDD-AGE assay in vitro using purified recombinant His-STING (residues 153–379) and GST-PTK2B expressed in *E. coli* showed that GST-PTK2B, but not GST-GFP (a negative control), increased STING oligomerization (Fig. 6b). Conversely, PTK2B deficiency in THP1 cells remarkably attenuated STING oligomerization induced by HSV1-GFP infection (Fig. 6c). Immunostaining assays showed that STING formed fewer and smaller granules in PTK2B-knockout THP1 cells stably expressing GFP-STING than it did in control cells upon HSV-1 infection (Fig. 6d). Moreover, we found that PTK2B deficiency remarkably reduced the formation of STING granules induced by cGAMP stimulation (Fig. 6e). These findings suggest that PTK2B increases STING oligomerization.

Next, we examined whether the kinase activity of PTK2B was required for its function in promoting STING oligomerization. First, we co-expressed STING with wild-type PTK2B or its kinase-inactive mutant PTK2B-K457R in HEK293T cells, and then performed in vivo kinase assays, and found that PTK2B increased STING tyrosine phosphorylation, whereas its kinase-inactive mutant did not have this effect (Fig. 6f). Surprisingly, however, we observed that both wild-type PTK2B and the PTK2B-K457R mutant promoted STING oligomerization (Fig. 6g). Consistently, immunostaining assays showed that both the wild-type and the kinase-inactive mutant of PTK2B augmented the formation of STING granules in HeLa cells (Fig. 6h). Collectively, these findings demonstrate that PTK2B can promote STING oligomerization in a kinase-independent manner.

## Discussion

TBK1 is a critical kinase in innate immune signaling and its activity must be properly controlled. However, how TBK1 activation is regulated remains incompletely understood. In the present study, we identified tyrosine kinase PTK2B as a TBK1-interacting protein, and the functional analyses in cells and in vivo demonstrated that PTK2B acted as a positive regulator of antiviral signaling. We also found that PTK2B phosphorylated TBK1 primarily at residue Y591, subsequently enhancing TBK1 oligomerization. In addition, we showed that PTK2B also interacted with STING and could augment its oligomerization in a kinase-independent mechanism. These data indicated that PTK2B increased the oligomerization of TBK1 and STING via different mechanisms. Together, the findings suggest that PTK2B mediates innate immune responses to viral infection by regulating the STING-TBK1 signaling.

TBK1 contains a scaffolding/dimerization domain at the C-terminal region. TBK1 dimerization or oligomerization is required for its kinase activity and autophosphorylation[21,37]. Previous studies showed that serine-threonine kinases were involved in regulating TBK1 activation, for example, GSK3β increased dimerization or oligomerization of TBK1 independent of its kinase activity[22], subsequently enhancing TBK1 activation, whereas AKT1 phosphorylated TBK1 at

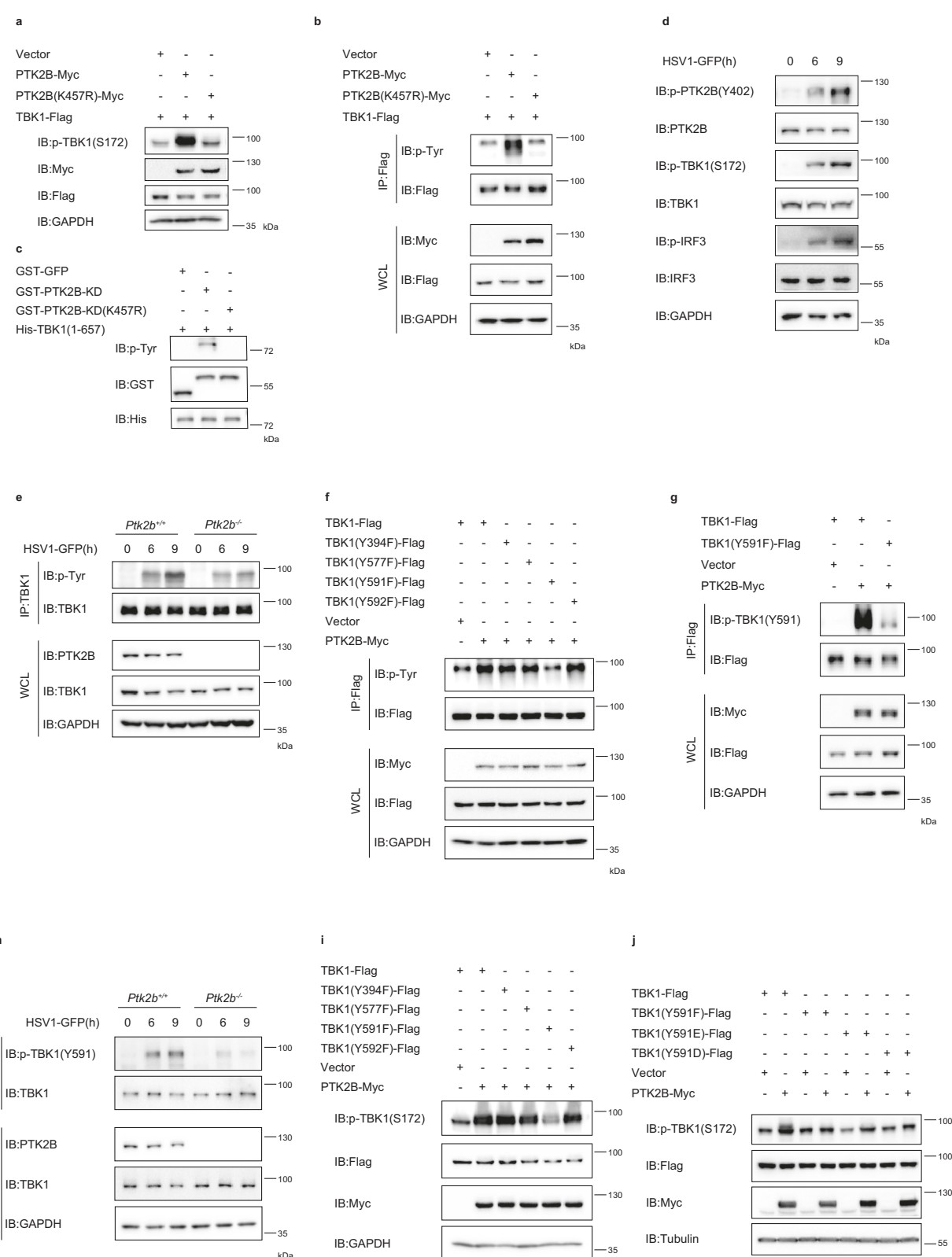

S510 to impair the association of TBK1 with STING or IRF3[13]. In addition, several tyrosine kinases were found to mediate TBK1 tyrosine phosphorylation and activation. LCK, HCK, and FGR have been shown to directly phosphorylate TBK1 at residues Y354 and Y394, and then suppress TBK1 dimerization and activation[16], whereas SRC was reported to increase TBK1 autophosphorylation and activation. However, in vitro kinase assays did not show that SRC directly phosphorylates TBK1[17]. How SRC regulated TBK1 activity remains unclear. Our immunostaining results showed that unlike the function of PTK2B, the knockdown of SRC had no effect on the formation of TBK1 granules induced by HSV1-GFP infection in immortalized MEFs (Supplementary Fig. S11a, b). These results suggest that PTK2B and SRC regulated TBK1 activation via different mechanisms. In the present study, we found that PTK2B was critical for the activation of TBK1 signaling in BMDMs,

**Fig. 4 | PTK2B augments TBK1 activation by mediating tyrosine phosphorylation. a** HEK293T cells were transfected with the indicated plasmids and lysed 24 h after transfection for immunoblotting. **b** HEK293T cells were co-transfected with Flag-tagged TBK1 and Myc-tagged PTK2B or PTK2B-K457R (a kinase-inactive variant). IP assays were performed with anti-Flag M2 beads and analyzed by immunoblotting. **c** Purified His-TBK1 protein (residues 1–657) was incubated with GST-PTK2B-KD or GST-PTK2B-KD (K457R) or GST-GFP for 30 min in kinase buffer, and then analyzed by immunoblotting. **d** RAW 264.7 cells were mock infected or infected with HSV1-GFP for the indicated times, followed by immunoblotting. **e** *Ptk2b*+/+ and *Ptk2b*−/− RAW 264.7 cells were mock infected or infected with HSV1-GFP for the indicated times. The cell lysates were immunoprecipitated with anti-TBK1 antibody and analyzed by immunoblotting. **f** HEK293T cells were co-transfected with Myc-tagged PTK2B and Flag-tagged TBK1 or its point mutants. IP assays were performed with anti-Flag M2 beads and pulled-down proteins were analyzed by immunoblotting. **g** HEK293T cells were co-transfected with Myc-tagged PTK2B and Flag-tagged TBK1 or TBK1-Y591F variant. IP assays were performed with anti-Flag M2 beads and pulled-down proteins were analyzed by immunoblotting. **h** *Ptk2b*+/+ and *Ptk2b*−/− RAW 264.7 cells were mock infected or infected with HSV1-GFP for 6 and 9 h. The cell lysates were immunoprecipitated with anti-TBK1 antibody and analyzed by immunoblotting. **i, j** HEK293T cells were co-transfected with the indicated plasmids and lysed 24 h after transfection for immunoblotting. Data shown in (**a**–**j**) are one representative of two independent experiments with similar results. Source data are provided as a Source Data file.

BMDCs and MEFs. Furthermore, we provided in vivo evidence that PTK2B was required for efficient defense against the infection of HSV-1 or VSV by employing Ptk2b-knockout mice. Mechanistically, we revealed that PTK2B directly phosphorylated TBK1 at residue Y591, which is located in its C-terminal scaffolding/dimerization domain, subsequently enhancing TBK1 activation by increasing its oligomerization. Given that PTK2B is highly expressed in immune cells, our findings suggest that PTK2B played an important role in activating sufficient immune responses to virus infection to ensure the efficient elimination of viruses.

PTK2 and PTK2B belong to the focal adhesion kinase family, and both are essential for NLRP3 inflammasome by targeting adapter protein ASC[27]. Although both PTK2 and PTK2B are involved in the formation of ASC oligomerization, only PTK2B directly phosphorylates ASC at residue Y146, leading to speck formation of ASC and subsequently inducing NLRP3 inflammation. How PTK2 modulated ASC oligomerization remains unclear. These findings indicated that PTK2 and PTK2B regulate NLRP3 inflammasome in different mechanisms. A previous study showed that PTK2 enhanced MAVS-mediated antiviral signaling in a kinase-independent manner[42]. In the present study, we found that PTK2, unlike PTK2B, did not phosphorylate TBK1, and overexpression or knockdown of PTK2 had no effects on the TBK1 oligomerization (Supplementary Fig. S11c, f). These findings suggest that PTK2 and PTK2B also employ different mechanisms to regulate antiviral signaling.

Previous studies have shown that STING can form dimers and undergo aggregation after binding to the second messenger, cGAMP, which is synthesized by cGAS[20,43,44]. We also found that upon HSV-1 infection, TBK1 and STING could co-localize with the PTK2B granules on Golgi in MEFs (Supplementary Fig. S10c). STING aggregation is critical for triggering the activation of TBK1-IRF3 signaling[11,23]. Our previous study showed that TBK1 promoted STING aggregation in a kinase activity-dependent manner, whereas phosphatase PPM1A reduced STING aggregation by dephosphorylating STING and TBK1[14]. Tyrosine phosphorylation of STING has been shown to play a role in regulating STING function. SRC was found to mediate STING activation by phosphorylating STING at residue Y245, but how SRC regulated STING activation is still unclear[45] Another tyrosine kinase SYK was shown to be required for STING translocation from ER to the ERGIC by phosphorylating STING at residue Y240[46]. In addition, one study indicated that the tyrosine-protein phosphatase PTPN1/2 dephosphorylated STING, and then promoted its 20S proteasomal degradation, subsequently reducing antiviral signaling[47]. In the present study, we found that overexpression of both wild-type PTK2B and its kinase-inactive mutant PTK2B(K457R) enhanced the aggregation of STING, and deficiency of PTK2B reduced the aggregation of STING stimulated by cGAMP and HSV1-GFP infection. These results indicated that PTK2B can augment STING aggregation in a kinase-independent manner.

Collectively, our results suggest that PTK2B regulates STING-TBK1 signaling by increasing the aggregation of TBK1 and STING in different mechanisms (Supplementary Fig. S12). These findings provide insight into the mechanisms by which PTK2B regulates STING-TBK1 activation

to ensure efficient antiviral responses and give a clue to the treatment of autoimmune diseases.

## Methods

### Ethics statements

All animal studies were carried out in strict accordance with the recommendations in the Guide for the Care and Use of Laboratory Animals of the Ministry of Science and Technology of the People's Republic of China. The protocols for animal studies were approved by the Committee on the Ethics of Animal Experiments of the Institute of Zoology, Chinese Academy of Sciences (approval number: IOZ15001).

### Cell culture and animals

HEK293T (GNHu17), HeLa (TCHu187), THP-1 (TCHu57), Vero (GNO10) and RAW 264.7 (TCM13) were kindly provided by Stem Cell Bank, Chinese Academy of Science from the Shanghai Cell Bank of the Chinese Academy of Sciences. HEK293T, HeLa, Vero, RAW 264.7 cells and immortalized MEFs and TBK1-deficient MEFs were maintained in Dulbecco's modified Eagle's medium (DMEM; Invitrogen) supplemented with 10% fetal bovine serum (Biological Industries), 1% penicillin and 1% streptomycin (Invitrogen). TBK1-knockout MEFs were kindly provided by Dr. Zhengfan Jiang's lab from Peking University and Immortalized MEFs from Dr. Tieshan Tang's lab from the Institute of Zoology. Primary MEFs were isolated from 13.5-day-old *Ptk2b*-deficient and littermate control embryos by crossing heterozygous mice and cultured in complete DMEM containing 1 mM sodium pyruvate, 10 mM L-glutamine, 10 mM β-mercaptoethanol and 1% nonessential amino acids (Invitrogen). THP-1 cells were maintained in RPMI-1640 medium (Invitrogen) supplemented with 10% fetal bovine serum, 1% penicillin, 1% streptomycin, 10 µM β-mercaptoethanol and 5 mM HEPES. For preparing bone marrow-derived macrophages, bone marrow cells were collected from femurs and tibias of mice and cultured in complete DMEM containing conditioned medium from a 5-day culture of L929 cells for 5 days, and then the cells were collected for experiments. Bone marrow-derived dendritic cells were prepared from bone marrow cells cultured with complete RPMI-1640 containing 10% supernatant from cultured B16-FLT3L cells for 7 days.

*Ptk2b* heterozygous mice (C57BL/6J) were obtained from Shanghai Model Organisms Center, Inc. Genomic DNA was isolated from 2-week-old mouse nails by One Step Mouse Genotyping Kit (Vazyme), followed by PCR analysis. All DNA oligonucleotides were synthesized by Tsingke Biological Technology Company. The sequence of primers for genotyping is as follows:

#1: 5'-ACCCGTCCGGATGAGAATAG3';
#2: 5'-CAGAACGAAAGCGATGGCG3'.

### Plasmids and transfection

For the construction of Flag-tagged or Myc-tagged mammalian expression plasmids, cDNA of PTK2B, cGAS, STING, TBK1 or IRF3 was amplified using 2× Phanta Max Master Mix (Vazyme) and inserted into pCDH, pcDNA3.0, pEF or pMXs vector. For the construction of bacterial expression plasmids for His-tagged STING (residues 153–379)

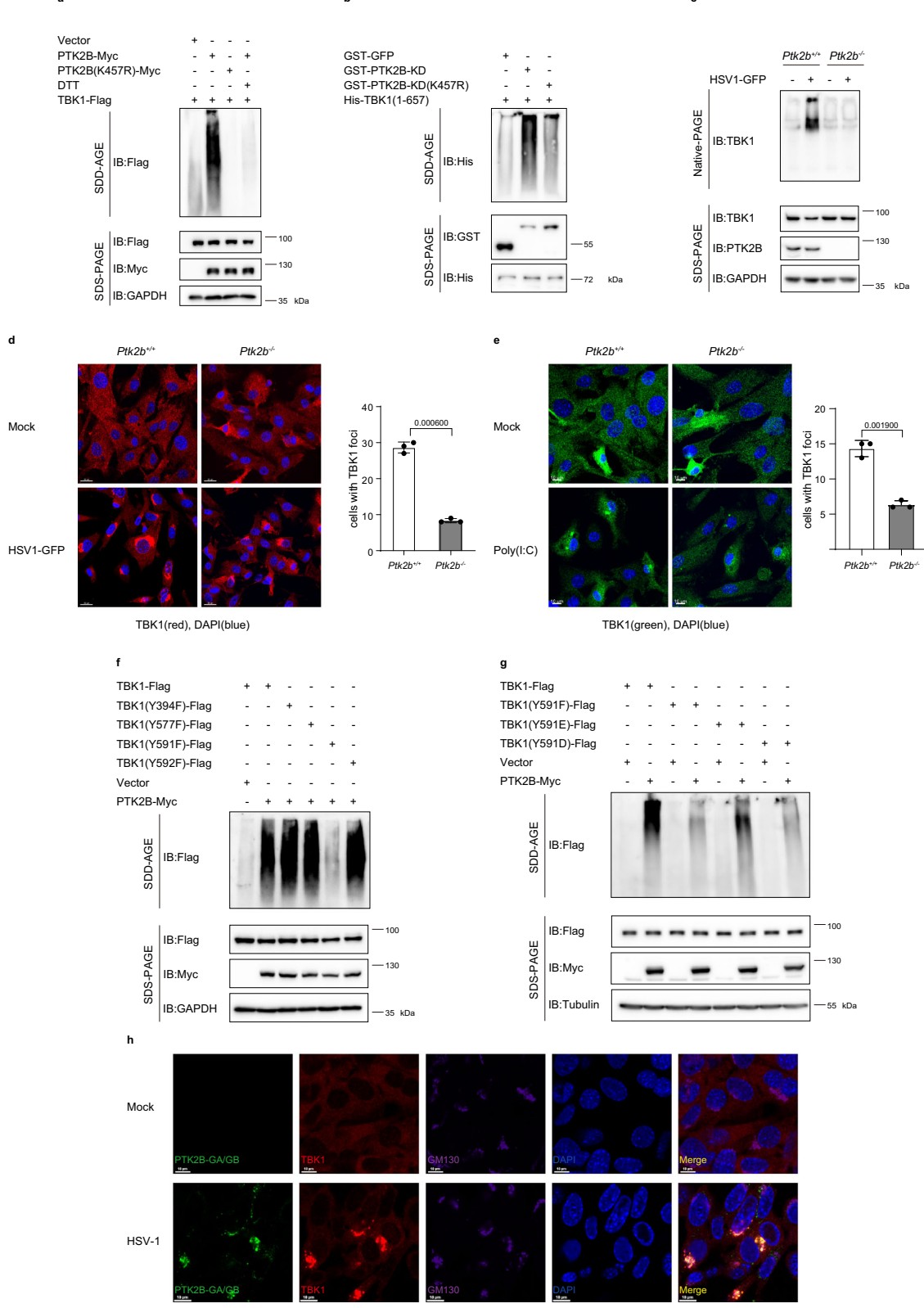

and TBK1 (residues 1–657) or GST-tagged PTK2B and PTK2B-KD (residues 360–690), cDNAs were subcloned into pET-28a or pET-30a, or pGEX4T-1vector, respectively. Truncation and point mutations were generated by PCR using *Pfu* DNA polymerase.

Plasmids were transfected into HEK293T or HeLa cells with polyethylenimine (Polysciences) or Lipo2000 (Invitrogen). HT-DNA was transfected with Lipo2000.

## Antibodies and reagents

Rabbit anti-STING (Cat# 50494, 1:1000), rabbit anti-human phospho-STING (Cat# 19781, 1:1000), rabbit anti-mouse phospho-STING (Cat# 72971, 1:1000), rabbit anti-IRF3 (Cat#4302, 1:500), rabbit anti-phospho-IRF3 (Cat# 4947, 1:500), rabbit anti-phospho-TBK1 (Ser172, 1:1000) (Cat# 5483), mouse anti-PTK2B antibodies (Cat# 3480, 1:1000), rabbit anti-phospho-PTK2B(Tyr402) antibodies (Cat# 3291,

**Fig. 5 | PTK2B increases TBK1 oligomerization in a kinase-dependent manner.**
**a** HEK293T cells were co-transfected with the indicated plasmids. Cell lysates were treated with or without 50 mM dithiothreitol (DTT) and then resolved by semi-denaturing detergent agarose gel electrophoresis (SDD-AGE) and sodium dodecyl sulfate-polyacrylamide gel electrophoresis (SDS-PAGE), followed by immunoblotting. **b** Purified His-TBK1 protein (residues 1–657) was incubated with GST-PTK2B-KD or GST-PTK2B-KD (K457R) or GST-GFP for 30 min, and then resolved by SDD-AGE or SDS-PAGE, followed by immunoblotting. **c** *Ptk2b*+/+ and *Ptk2b*−/− RAW 264.7 cells were mock infected or infected with HSV1-GFP for 6 h. The cell lysates were resolved by Native-PAGE or SDS-PAGE, followed by immunoblotting. **d, e** Primary *Ptk2b*+/+ and *Ptk2b*−/− MEFs were stimulated by HSV1-GFP infection for 6 h (**d**) or poly(I:C) transfection for 2 h (**e**), stained with TBK1 antibody and 4′,6-diamidino-2-phenylindole (DAPI), and then imaged by confocal microscopy (left). Scale bars in

(**d**) 20 μm and in (**e**) 10 μm. The percentage of cells with TBK1 foci was quantified (right, *n* = 108 cells in (**d**) and *n* = 103 cells in (**e**) for each group). **f** HEK293T cells were co-transfected with the indicated plasmids. Cell lysates were resolved by SDD-AGE and SDS-PAGE, followed by immunoblotting with the indicated antibodies. **g** Similar to (**f**), except that the indicated expression plasmids were transfected. **h** Immortalized MEFs were co-expressed with PTK2B-GA and PTK2B-GB, followed by stimulation with or without HSV-1 for 6 h. The cells were fixed and stained with DAPI (blue), antibody to TBK1 (red) and GM130 (Golgi marker, purple), then imaged by confocal microscopy. Scale bars, 10 μm. Data shown in (**a–c**, **f–h**) are one representative of two independent experiments with similar results. Data shown in (**d, e**) are from one representative experiment of three independent experiments (mean ± SD), two-tailed Student's *t*-test. Source data are provided as a Source Data file.

1:1000), mouse anti-PDI (ER marker) antibodies (Cat# 45596, 1:300), mouse anti-phospho-Tyrosine antibodies (Cat# 9416, 1:1000) and rabbit anti-Rig-I (Cat# 3743, 1:500) were from Cell Signaling Technology. Rabbit anti-TBK1 (Cat# ab40676, 1:5000) and rabbit anti-PTK2B (Cat# ab226798, 1:1000) antibodies were from Abcam. Rabbit anti-TBK1 (Cat# sc-52957, 1:1000) antibody was from Santa Cruz Biotechnology. Rabbit anti-STING for immunostaining (Cat# 19851-1-AP, 1:1000) was from Proteintech. Mouse anti-GM130 (Golgi marker, Cat#610822, 1:1000) was from BD Biosciences. Mouse anti-glyceraldehyde-3-phosphate dehydrogenase (GAPDH) (Cat# KM9002, 1:5000) and mouse anti-α-Tubulin (Cat# KM9007, 1:5000) antibodies were from Sungene Biotechnology. Mouse anti-Flag (Cat# M185-3, 1:5000), rabbit anti-HA (Cat# 561, 1:5000) and rabbit anti-Myc (Cat# 562, 1:5000) antibodies were from MBL. Mouse anti-GST antibody (Cat# TA150101, 1:1000) was from OriGene Technologies, Inc. Rabbit anti-His antibody (Cat# B1023, 1:1000) antibody was from Biodragon. Specific antibody for the phosphorylation of TBK1 at tyrosine 591(1:1000) was generated in ABclonal Technology. Goat anti-Rabbit IgG HL(HRP)(Abcam, Cat# AB205718, 1:5000) and Goat anti-mouse IgG HL(HRP) (Abcam, Cat# AB205719, 1:5000) were from Abcam. Goat anti-Mouse IgG (H + L) Cross-Adsorbed Secondary Antibody, Alexa Fluor™ 488(A-11001, 1:2000), Goat anti-Rabbit IgG (H + L) Cross-Adsorbed Secondary Antibody, Alexa Fluor™ 555(A-21428, 1:2000), Goat anti-Rabbit IgG (H + L) Cross-Adsorbed Secondary Antibody, Alexa Fluor™ 488(A-11008, 1:2000), Goat anti-Mouse IgG (H + L) Cross-Adsorbed Secondary Antibody, Alexa Fluor™ 647(A-21235, 1:2000), Goat anti-Mouse IgG (H + L) Cross-Adsorbed Secondary Antibody, Alexa Fluor™ 555(A-21422, 1:2000), Goat anti-Rat IgG (H + L) Cross-Adsorbed Secondary Antibody, Alexa Fluor™ 647(A-21247, 1:2000) and Goat anti-Mouse IgG (H + L) Cross-Adsorbed Secondary Antibody, Alexa Fluor™ 405(A-31553, 1:2000) were from Invitrogen. Rat anti-mouse/human CD11b brilliant violet 510 (Cat# 101263, 1:50), Rat anti-mouse F4/80 APC (Cat# 123115, 1:125), Rat anti-mouse Ly-6G PE (Cat# 127607, 1:125) and American hamster anti-mouse CD11c (Cat# 117305, 1:500) were from Biolegend. All antibodies are used according to the instructions of the product datasheet.

## Co-IP, sodium dodecyl sulfate-polyacrylamide gel electrophoresis (SDS-PAGE) and immunoblotting analysis
For in vivo exogenous Co-IP assays, transfected cells were lysed in lysis buffer (20 mM Tris-HCl, pH 7.5, 150 mM NaCl, 0.5% Triton X-100, 10% glycerol, 1 mM ethylenediaminetetraacetic acid) on ice supplemented with a complete protease inhibitor cocktail (Roche, cat# 4693132001) or PhosSTOP EASYpack (Roche, cat# 4906837001). After centrifugation at 16,000×*g* for 10 min at 4 °C, clarified cell lysates were incubated with anti-Flag M2 agarose beads (Sigma-Aldrich, cat# A2220) for 4 h at 4 °C. The immunoprecipitated complexes were washed with lysis buffer containing 150 mM NaCl three times and subjected to immunoblotting. For endogenous Co-IP, cell lysates were incubated with PTK2B or TBK1 or STING antibody or IgG antibody as a control, overnight at 4 °C, followed by incubation with protein A/G beads (Pierce,

cat# 53133) for 2 h. Immunoprecipitated complexes were washed with lysis buffer containing 150 mM NaCl three times and subjected to SDS-PAGE and immunoblotting with the indicated antibodies. The software used for collecting immunoblotting data was FLICapture (version 1.02). For the detailed SDS-PAGE assay, in brief, immunoprecipitated complexes or clarified cell lysates were mixed in 1× SDS loading buffer (50 mM Tris-Cl, pH 6.8, 50 mM dithiothreitol, 2% SDS, 0.05% bromophenol blue, and 10% glycerol), heated for 10 min at 95°C, and loaded onto a vertical 10% polyacrylamide gel. Electrophoresis was performed in 1× SDS running buffer (25 mM Tris, pH 6.8, 250 mM Glycine, 0.1% SDS) at a constant 90–130 V for 1 h. Uncropped and unprocessed scans of blots in figures and supplementary figures are supplied in the Source Data file or at the end of the supplementary figures in the Supplementary Information, respectively.

## Purification of recombinant proteins from *E. coli* and in vitro pull-down assays
Plasmids pGEX-4T-1-GST-PTK2B, pGEX-4T-1-GST-PTK2B-KD (residues 360–690) pGEX-4T-1-GST-PTK2B-KD(K457R), pGEX-4T-1-GST-GFP, pET28a-His-STING (residues 153–379), pET30a-His-TBK1 (residues 1–657) and pET28a-His-GFP were respectively transformed into *E. coli* BL21. The fusion proteins were purified from cell lysates using Ni-Sepharose beads (GE Healthcare, cat# 17-5318-02) or Glutathione-Sepharose beads (GE Healthcare, cat# 17-0756-01) in accordance with the manufacturer's protocols.

For GST pull-down assays, GST-tagged PTK2B protein or GFP was incubated with recombinant His-STING protein (residues 153–379) at 4 °C for 30 min and then incubated with Glutathione-Sepharose 4B at 4 °C for 4 h in GST-binding/washing buffer [43 mM $Na_2HPO_4$, 14.7 mM $KH_2PO_4$, 1.37 mM NaCl, 27 mM KCl, 0.1% Triton X-100 and 1 mM phenylmethylsulfonyl fluoride (PMSF)]. The immunoprecipitated complexes were washed three times and subjected to immunoblotting.

For His pull-down assays, His-tagged TBK1 protein (residues 1–657) or GFP was incubated with recombinant GST-PTK2B protein at 4 °C for 30 min and then incubated with Ni-Sepharose beads at 4 °C for 4 h in His-binding buffer (20 mM Tris-HCl, pH 8.0, 500 mM NaCl, 10 mM imidazole, 1 mM PMSF). The immunoprecipitated complexes were washed with His-washing buffer (20 mM Tris-HCl, pH 8.0, 500 mM NaCl, 20 mM imidazole, 1 mM PMSF) three times and subjected to immunoblotting.

## Semi-denaturing detergent agarose gel electrophoresis (SDD-AGE)
For the in vivo SDD-AGE assay, the transfected cells were harvested in lysis buffer (0.5% Triton X-100, 50 mM Tris-HCl, 150 mM NaCl, 10% glycerol), sonicated and then incubated on ice for 30 min. After centrifugation at 16,000×*g* for 10 min at 4 °C, clarified cell lysates were mixed in 1× SDD loading buffer [0.5× TBE, 10% glycerol, 2% sodium dodecyl sulfate (SDS), 0.0025% bromophenol blue] and loaded onto a vertical 1.5% agarose gel (1× TBE). Electrophoresis was performed in 1× TBE running buffer (1× TBE, 0.1% SDS) for 35 min at 4 °C at a constant

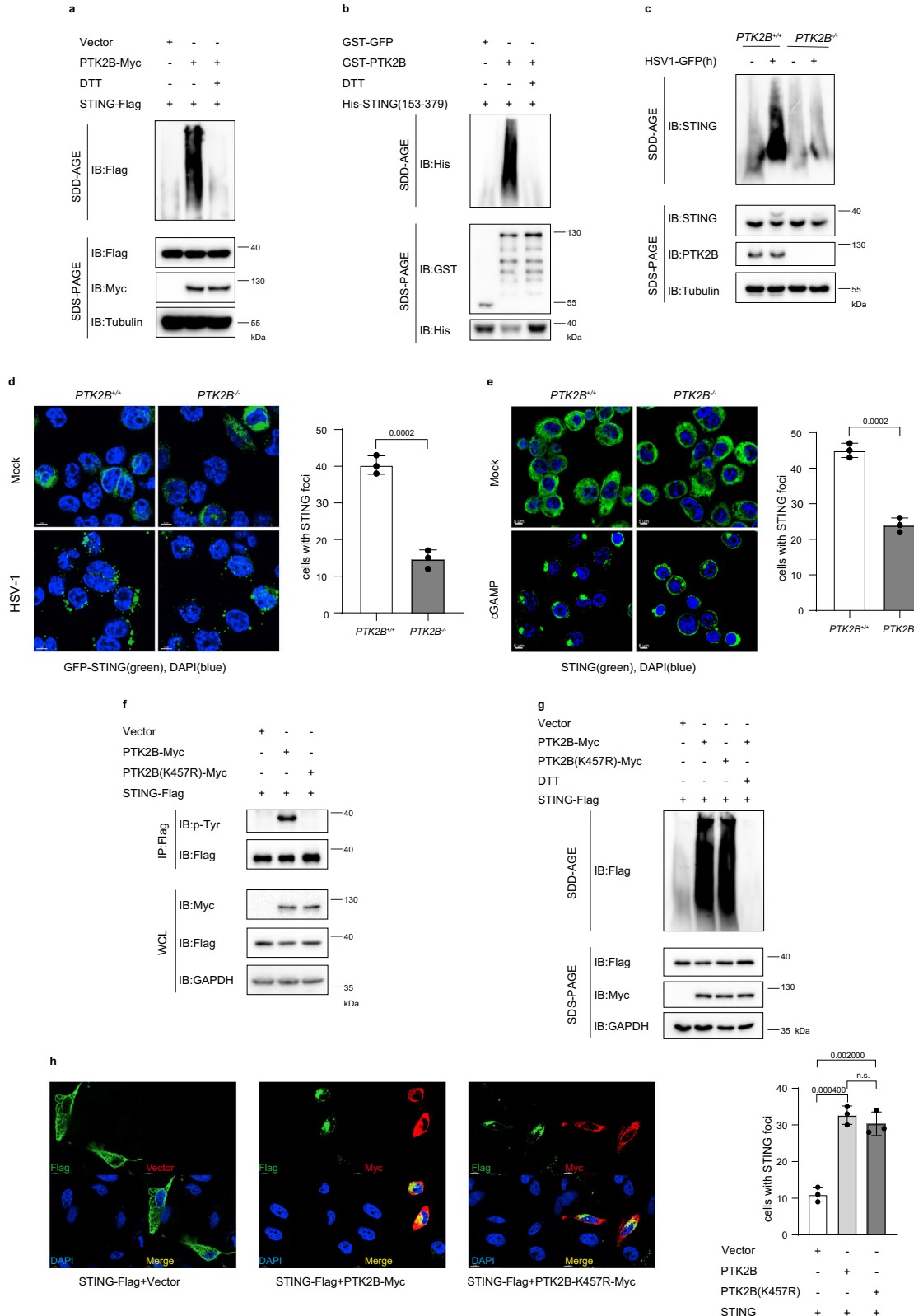

100 V, followed by immunoblotting. For in vitro aggregate detection, mixed equal amounts (0.5 μg) of recombinant TBK1 or STING (residues 153–379) protein, and recombinant PTK2B protein or its kinase-inactive mutant, were incubated in low-salt buffer (25 mM Tris-HCl, pH 7.5, 10 mM MgCl₂, 200 μM ATP) at 30 °C for 30 min. The reactions were terminated by adding SDD loading buffer and the samples were resolved by SDD-AGE.

## Native polyacrylamide gel electrophoresis (Native-PAGE)

The gels were pre-run with running buffer (25 mM Tris-Cl, pH 8.4, 190 mM glycine, in the presence and absence of 0.2% deoxycholate in the cathode and anode buffer, respectively) at 200 V for 1 h at 4 °C. Whole cells were collected, lysed, sonicated and centrifuged as described above for SDD-AGE assays. The samples were resolved in 1× Native loading buffer (62.5 mM Tris-Cl, pH 6.8, 10% glycerol, 0.4%

**Fig. 6 | PTK2B can enhance STING oligomerization in a kinase-independent manner. a** HEK293T cells were co-transfected with Flag-STING and Myc-PTK2B or empty vector. Cell lysates were incubated in the presence or absence of 50 mM DTT for 30 min, and then resolved by SDD-AGE and SDS-PAGE, followed by immunoblotting. **b** Purified His-STING protein (residues 153–379) was incubated with GST-PTK2B or GST-GFP in the presence or absence of 50 mM DTT for 30 min, and then resolved by SDD-AGE or SDS-PAGE, followed by immunoblotting. **c** *PTK2B*$^{+/+}$ and *PTK2B*$^{-/-}$ THP1 cells were infected with HSV1-GFP for 10 h. Cell lysates were resolved by SDD-AGE and SDS-PAGE, followed by immunoblotting. **d** *PTK2B*$^{+/+}$ and *PTK2B*$^{-/-}$ THP1 cells stably expressing GFP-STING were infected with HSV-1 infection for 10 h and then imaged by confocal microscopy (left). Scale bars, 5 μm. The percentage of cells with STING foci was quantified (right, *n* = 102 cells for each group). **e** *PTK2B*$^{+/+}$ and *PTK2B*$^{-/-}$ THP1 cells were stimulated with cGAMP (1 μM) for 1.5 h, then imaged by confocal microscopy (left). Scale bars, 5 μm. The percentage of cells with STING foci was quantified (right, *n* = 105 cells for each group). **f** HEK293T cells were co-transfected with Flag-tagged STING and Myc-tagged PTK2B, PTK2B-K457R variant, or empty vector. Cell lysates were immunoprecipitated with anti-Flag M2 beads, and then the pulled-down proteins were analyzed by immunoblotting. **g** Similar to (**a**), except that PTK2B-K457R expression plasmid was included. **h** HeLa cells were co-transfected with Flag-tagged STING and Myc-tagged PTK2B, PTK2B-K457R, or empty vector. Twenty-four hours after transfection, the cells were stained with Flag and Myc antibodies and then imaged by confocal microscopy (left). Scale bars, 10 μm. The percentage of cells with STING foci was quantified (right, *n* = 103 cells for each group). Data shown in (**a–c**, **f**, **g**) are one representative of two independent experiments with similar results. Data shown in (**d**, **e**, **h**) are from one representative experiment of three independent experiments (mean ± SD), n.s. not significant, two-tailed Student's *t*-test. Source data are provided as a Source Data file.

deoxycholate, 0.0025% bromophenol blue), electrophoresed at 200 V for 1 h at 4 °C, and subjected to immunoblotting.

### Immunofluorescence assays

HeLa, MEFs, or THP-1 cells were seeded on gelatin-coated glass coverslips in 12-well plates and then transfected or stimulated with the indicated treatment. The cells were washed with phosphate-buffered saline (PBS), fixed with 4% paraformaldehyde for 15 min, washed with PBS once and permeabilized with 0.2% Triton X-100 for 10 min, and blocked with 5% (w/v) bovine serum albumin for 30 min at room temperature. Then, the cells were incubated with primary and secondary antibodies. The cells were washed with PBST (PBS with 0.2% Tween 20) between each step. Imaging was performed with an Andor Dragonfly 505 laser scanning system, and the images were analyzed using Imaris 9.5.1 software. For quantitative co-localization analysis, images were acquired with a Nikon A1 confocal microscope. Pearson's correlation coefficients were determined using NIS-Elements AR analysis 5.20.00 software.

### Generation of stable cell lines using lentivirus or retrovirus

Full-length cDNA encoding mouse Ptk2b or Tbk1 was amplified and inserted into expression vector pCDH-CMV-Puro. Lentivirus particles were produced by co-transfecting pCDH-CMV-Puro-Ptk2b or pCDH-CMV-Puro-Tbk1 into HEK293T cells with the packaging plasmids pMD2.G and psPAX2. The indicated cells were infected with lentiviral particles expressing PTK2B, TBK1 or empty vector for 48 h, followed by subsequent experiments. Full-length cDNA encoding mouse TBK1 was amplified and inserted into Retrovirus expression vector pMXs. Retrovirus expressing TBK1 was produced by co-transfecting pMXs-TBK1 into HEK293T cells with the packaging plasmid Ecopac for the subsequent experiments.

To generate PTK2B-knockdown cells, we employed pLKO.1-puro-based lentivirus expressing specific shRNAs against the target gene. THP-1, immortalized MEFs or RAW264.7 cells were infected with lentivirus targeting *PTK2B, Ptk2, Src* or negative control shRNA (shCon), and then selected with puromycin. The knockdown efficiency was determined by qPCR or western blotting analysis. The sequences (5′–3′) against the target gene for shRNAs were as follows.

Human PTK2B shRNA-1#: TCTGTCCACCGGTCCTTTATA;
Human PTK2B shRNA-2#: AGAACTTCAAACTGGTCAAAT;
Mouse Ptk2b shRNA: CGCATCCTCAAGGTCTGCTTC;
Mouse Ptk2 shRNA: CCTGGCATCTTTGATATTATA;
Mouse Src shRNA: GCGGCTGCAGATTGTCAATAA

To generate stable PTK2B knockout cells, the corresponding single guide RNAs (sgRNAs) were inserted into the LentiCRISPRv2 vector according to the method described by Sanjana et al.[35] The cells were infected with lentivirus containing sgRNAs targeting PT2KB and selected with puromycin for at least 6 days. PTK2B-knockout cells were verified by immunoblotting. PTK2B sgRNA sequences (5′–3′) were as follows:

Human PTK2B-sgRNA1: TCTACTGGTACCACCACCAT;
Human PTK2B-sgRNA2: CTTTACTCGACTCAGGGGCT;
Mouse Ptk2b-sgRNA1: GCCCTTGAGCCGTGTAAAAG;
Mouse Ptk2b-sgRNA2: GGAAGACGTGCGCATCCTCA.

### qPCR

Total RNA was extracted from cells using TRIzol reagent (Invitrogen) and cDNA was synthesized using a HiScript III First-Strand cDNA Synthesis kit (Vazyme). The qPCRs were performed using SYBR Green Master Mix (Thermo Fisher) in a CFX96 Optics Module (Bio-Rad). The $2^{-\Delta\Delta Ct}$ method was used to calculate relative levels of gene expression, and the relative mRNA level for each gene was normalized to the mRNA level of GAPDH. Data are shown as the relative abundance of the mRNA compared with that of the control groups. All samples were assayed in triplicate. The gene-specific primers were as follows (5′–3′):

Human PTK2B-S: AGATGTGGAAAAGGAGGACG;
Human PTK2B-AS: CTCAGCAGGATGGAGGTGAT;
Human IFNB1-S: AGGACAGGATGAACTTTGAC;
Human IFNB1-AS: TGATAGACATTAGCCAGGAG;
Human IFIT1-S: TACCTGGACAAGGTGGAGAA;
Human IFIT1-AS: GTGAGGACATGTTGGCTAGA;
Human CXCL10-S: GGTGAGAAGAGATGTCTGAATCC;
Human CXCL10-AS: GTCCATCCTTGGAAGCACTGCA;
Human GAPDH-S: ATGACATCAAGAAGGTGGTG;
Human GAPDH-AS: CATACCAGGAAATGAGCTTG;
Mouse Gapdh-S: AACTTTGGCATTGTGGAAGG;
Mouse Gapdh-AS: ACACATTGGGGGTAGGAACA;
Mouse Ifnb1-S: ATGGTGGTCCGAGCAGAGAT;
Mouse Ifnb1-AS: CCACCACTCATTCTGAGGCA;
Mouse Ifit1-S: CTGAGATGTCACTTCACATGGAA;
Mouse Ifit1-AS: GTGCATCCCCAATGGGTTCT;
Mouse Cxcl10-S: ACTGCATCCATATCGATGAC;
Mouse Cxcl10-AS: TTCATCGTGGCAATGATCTC;
Mouse Ptk2b-S: CAGATGACCGTGGGCGAAGT;
Mouse Ptk2b-AS: GGCAAGTAGCGGATTTGAAGG
Mouse Ptk2-S: CAAAAGGATTTCTAAACCAG;
Mouse Ptk2-AS: TTTCTTGCATGTAGTCACTCT;
Mouse Src-S: TGGACAGCGGCGGTTTCTAC;
Mouse Src-AS: GTCTGAGGCTTGGATGTGGG.

### Viral plaque assay

*Ptk2b*$^{+/+}$ and *Ptk2b*$^{-/-}$ RAW 264.7 cells were infected with HSV1-GFP [multiplicity of infection (MOI) 0.5] for 24 h or VSVΔM51-GFP (MOI 0.1) for 12 h. The cells were imaged by fluorescence microscopy, or the culture supernatants were harvested to quantify the viral titer by a plaque assay. Culture supernatants were collected and diluted to infect Vero cells; After 72 h (HSV1-GFP) or 48 h (VSVΔM51-GFP), Vero cells were fixed with methanol for 30 min and stained with 1% crystal violet. Plaques were counted to quantify viral titer, shown in plaque-forming units (pfu)/mL.

## Viral infection in mice and measurement of viral titer

*Ptk2b*[+/+] and *Ptk2b*[−/−] mice were infected with HSV-1 ($4 \times 10^7$ pfu per mouse) or VSV ($1 \times 10^8$ pfu per mouse) via tail vein injection. Sera were collected after infection to measure the levels of IFNβ, IL-6 and TNFα using an ELISA kit (PBL Biomedical Laboratories) following the instructions of the manufacturer (OPTIMA, version 2.20) or viral titers. The viability of the infected mice was monitored for 15 days. In addition, *Ptk2b*[+/+] and *Ptk2b*[−/−] mice were infected with HSV1 via tail vein injection at $3 \times 10^7$ pfu per mouse for 4 days, and then the infected mice were sacrificed and lung tissues were harvested to detect inflammation by hematoxylin and eosin staining or perform flow cytometry.

## Identification of TBK1-interacting proteins by mass spectrometry

HEK293T cells were transfected with SFB-tagged mTBK1 or an empty vector for 24 h, the cells were lysed with lysis buffer (20 mM Tris-HCl, pH 7.5, 150 mM NaCl, 0.5% Triton X-100, 10% glycerol, 1 mM ethylenediaminetetraacetic acid) containing a complete protease inhibitor cocktail, followed by centrifugation at 16,000×*g* for 10 min at 4 °C. The supernatants were subjected to immunoprecipitation using S-protein agarose (Millipore, Cat# 69704) for 3 h at 4 °C. Immunoprecipitates were washed with lysis buffer containing 150 mM NaCl three times and incubated with cell lysates from RAW 264.7 cells overnight at 4 °C. The immunoprecipitated complexes were washed three times again and separated by SDS-PAGE, and the gel was stained with Coomassie brilliant blue. The entire lane was cut into 2-mm gel slices, digested with Trypsin, and subjected to LC-MS/MS assays using an Orbitrap Elite mass spectrometer (Thermo Fisher Scientific). The mass spectrometry data were analyzed using Thermo Proteome Discovery (Version 2.3), and tandem mass spectra were searched against the UniProt- the UniProt Mus musculus database 2022.11.16.

## Dimerization-dependent FP (ddFP) assay

To detect the behavior of PTK2B protein in cells, we employed a dimerization-dependent fluorescent proteins (ddFP) assay system as described previously[48,49]. The ddFP assay system involves a pair of reversible dark FP monomers (GA and GB) fusion with the interested proteins to form a fluorescent heterodimeric complex. We first generated the lentiviral expression plasmids containing PTK2B-GS linker-GA (GA: Addgene#61018) and PTK2B-GS linker-GB (GB: Addgene#61017). The GS linker is GGAAGCGGAAGCGGA. Immortalized MEF cells were infected with lentivirus expressing PTK2B-GA and PTK2B-GB and then infected with HSV-1 for 6 h. Images were acquired with a Nikon A1 confocal microscope.

## Antisense oligonucleotides (ASO)

In order to downregulate the PTK2B expression, we designed a series of intron or exon ASOs targeted to the transcripts of the *PTK2B* gene. All the gapmer (5-10-5 MOE-gapmer) ASOs have five 2′-O-methoxyethyl (2′-MOE) modified nucleotides on either end which can enhance the nuclease stability and increase the affinity for target RNAs, ten DNA nucleotides in the center which enable RNase H1 to digest the hybrid DNA-RNA. All the ASOs are fully modified with phosphorothioate internucleotide linkages which can improve nuclease resistance and the binding to plasma proteins[50]. All the ASOs were transfected to A549 cells with the Celopener transfection reagent (GeneBio). Total RNA was extracted after 24 or 48 h transfection for analysis of mRNA levels of *PTK2B* by qPCR. ASO sequences and modification patterns are listed in Supplementary Table S2.

## Flow cytometric analysis and antibodies

*Ptk2b*[+/+] and *Ptk2b*[−/−] mice were infected with HSV-1 via tail vein injection at $3 \times 10^7$ pfu per mouse for 4 days. The cells were prepared by digesting mouse lung tissue with collagenase at 37 °C for 30 min and incubated with the designated antibody for 30 min at 4 °C. After washing three times with 1×PBS, A BD LSRFortessa X-20 flow cytometer (BD Biosciences, USA) was used for the data collection and FlowJo (version X.0.7) was used for the data analysis.

The following antibodies were used: Rat anti-mouse/human CD11b brilliant violet 510™ (Biolegend, Cat# 101263), Rat anti-mouse F4/80 APC (Biolegend, Cat# 123115), Rat anti-mouse Ly-6G PE (Biolegend, Cat# 127607), American hamster anti-mouse CD11c (Biolegend, Cat# 117305).

## Phosphorylation site identification by mass spectrometry

HEK293T cells were co-transfected with Flag-tagged TBK1 and Myc-tagged PTK2B or empty vector. Cell lysates were immunoprecipitated with anti-Flag M2 beads. Tryptic digestion, desalting over C18 and enrichment of phosphopeptides over titanium dioxide ($TiO_2$) beads were performed according to the method described by Wu et al.[51]. In brief, immunoprecipitated proteins collected using anti-Flag M2 beads were predigested for 3 h with endoproteinase Lys-C (0.5 µg/µL; Wako Chemicals, Neuss, Germany) at room temperature. After four times dilution with 10 mM Tris-HCl (pH 8.0), samples were then digested with sequencing-grade modified trypsin (0.5 µg/µL; Promega) at 37 °C overnight. Digested peptides were desalted over a C18 stage tip[52] and dissolved in 1 M glycolic acid in 80% (v/v) acetonitrile and 6% (v/v) trifluoroacetic acid (TFA) before phosphopeptide enrichment with $TiO_2$ (GL Sciences). Dissolved peptides were mixed with $TiO_2$ beads for 30 min, then the $TiO_2$ bead and peptide mixture was washed twice with 80% (v/v) acetonitrile and 1% (v/v) TFA. Phosphopeptides were eluted from $TiO_2$ beads three times with 1% (v/v) ammonia solution. The eluates were acidified and desalted over a C18 stage tip.

Phosphopeptide mixtures were analyzed using a nanoflow Easy-nLC (Thermo Scientific) and an Orbitrap hybrid mass spectrometer (Orbitrap Exploris 480, Thermo Scientific). Peptides were eluted from a 75-µm, 25-cm home-made analytical C18 column with a linear gradient from 5% to 90% acetonitrile over 135 min. Proteins were identified based on the information-dependent acquisition of fragmentation spectra of multiple charged peptides. Data-dependent MS/MS spectra were acquired in the linear ion trap for each full-scan spectrum acquired at 70,000 full-width half-maximum resolution.

MaxQuant software version 1.6.4.0[53] was used for raw file peak extraction and protein identification against the UniProt *Homo sapiens* database UP000005640 (80,581 entries). The following parameters were applied: trypsin as cleaving enzyme; minimum peptide length of seven amino acids; maximal two missed cleavages; carbamidomethylation of cysteine as a fixed modification; oxidation of methionine and phosphorylation of serine, threonine and tyrosine as variable modifications. Peptide mass tolerance was set to 20 ppm and 0.5 Da was used as the MS/MS tolerance. Further settings were peptide and protein false discovery rates set to 0.01; common contaminants (trypsin, keratin, and so on) were excluded.

## Statistics and reproducibility

The log-rank (Mantel–Cox) test was used for survival rate analysis. Other statistical analyses are shown as mean ± SD. Statistical analysis was carried out using the two-tailed Student's *t*-test (GraphPad Prism 8.0). For all tests, *P*-values of <0.05 were considered statistically significant.

## Reporting summary

Further information on research design is available in the Nature Portfolio Reporting Summary linked to this article.

## Data availability

The mass spectrometry proteomics data have been deposited to the ProteomeXchange Consortium (http://proteomecentral.proteomexchange.org) via the iProX partner repository[54,55] with the dataset identifier PXD045961. Source data are provided with this paper.

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

## Acknowledgements

We would like to thank Dr. Daxing Gao (University of Science and Technology of China), Dr. Zhengfan Jang (Peking University) and Dr. Tieshan Tang (Institute of Zoology, CAS,) for kindly providing HSV-1 and HSV1-GFP viruses, TBK1-deficient MEFs and immortalized MEFs, respectively. We also like to thank Dr. Xuna Wu (The proteomics facility of Life Science College, Yunnan University) and Dr. Kehui Liu (The mass spectrometry platforms in State Key Laboratory of Membrane Biology, Institute of Zoology) for mass spectrometry assays. This work is supported by the Natural Science Foundation of China (Grant 32370928, 31970895, Q.S.), the Basic Science Center Program of NSFC (Grant 31988101, D.C.) and the Open Research Program of State Key Laboratory of Membrane Biology.

## Author contributions

Q.S., Y.L., and D.C. conceived this study. Q.S., Y.L., and D.C. designed the experiments. Y.L., J.Y., Q.Y., J.Z., S.Z., Y.Z., Y.T., and L.L. conducted the experiments. Q.S., Y.L., J.Y., Q.Y. and W.T. analyzed the data. Q.S., Y.L., and D.C. wrote the manuscript. All authors provided intellectual input, vetted and approved the final manuscript.

## Competing interests

The authors declare no competing interests.
