## [Peer Review File · Nature Communications]

PTK2B promotes TBK1 and STING oligomerization and enhances the STING-TBK1 signalingREVIEWER COMMENTS

Reviewer #1 (Remarks to the Author):

In this manuscript, the authors demonstrated that protein tyrosine kinase 2 beta (PTK2B) (also known as PYK2 or FAK2) is a positive regulator of STING-TBK1 activation in antiviral innate immune responses. They showed that PTK2B interacts with TBK1 and STING, and PTK2B positively regulates expression of antiviral genes. Ptk2b-deficient mice were more susceptible to HSV-1 viral infection than control mice. Mechanistically, they showed that PTK2B promotes TBK1 activation and oligomerization by TBK1 phosphorylation at Y591. Additionally, PTK2B enhances STING oligomerization in a kinase-independent manner.

Although the role of PTK2B in the activation of TBK1 is potentially interesting, the importance of this protein in antiviral immunity by type I IFN stimulation looks to be limited and the analysis of the roles of PTK2B in antiviral innate immune signaling is still preliminary. Further, although the authors identified several phosphorylation sites of TBK1 by PTK2B, how each phosphorylation is important under virus infection is unclear. Specific comments are shown below.

Major points

1. Although the authors show that Ptk2b KO mice are highly susceptible to HSV-1 infection, the increase in viral titer and the decrease of IFN β production was quite modest in vivo (Fig. 3). It is unlikely that such a modest impairment in IFN β production leads to the huge difference in the lethality of mice. As the authors stated, PTK2B has been reported to be involved in the activation of inflammasome (Sci Rep 2016). Thus, the authors need to clarify if the inflammasome contributes to the alteration in the lethality of HSV-1-infected Ptk2b KO mice.
2. Fig. 3I showed that infiltration of immune cells and tissue damage were increased in the lungs of infected Ptk2b-KO mice than in WT mice. However, it is unclear what kinds of immune cells are infiltrated in lungs and whether population and activation status of infiltrated immune cells are different between Ptk2b WT and KO mice.
3. It is also important to investigate the survival of Ptk2b KO mice in response to the infection with RNA viruses to further tighten the relationship of PTK2B with the activation of TBK1.
4. It is also essential to uncover the mechanisms how PTK2B is activated in response to virus infection to phosphorylate TBK1 and induce STING oligomerization.
5. Fig. 1 showed that PTK2B interacts with TBK1 and STING. The authors should check if PTK2B colocalizes with TBK1 granules after virus infection more clearly. Unfortunately, the scale of the images in Fig. 1d is not suitable to tell this. Additionally, it should be shown if PTK2B localizes to ER together with TBK1 and STING upon viral infection.
6. The authors should also check the subcellular localization of TBK1-Y591F. Does TBK1-Y591F form granules after virus infection?
7. The authors showed that PTK2B binds to STING as well as TBK1. Since PTK2B is also involved in IFN induction under RNA virus infection (Fig. S2 and S3), they should check if PTK2B binds to MAVS or other proteins involved in RNA virus sensing.
8. Fig. 2f-h and S3a-d showed that PTK2B overexpression increased the mRNA levels of antiviral genes upon infection with HSV1-GFP and VSV Δ M51-GFP in MEFs. It should be shown whether the levels of phosphorylated STING, TBK1, and IRF3 are increased by PTK2B overexpression.
9. In Fig. 4a-b and 5a-b, the authors showed that PTK2B kinase activity is required for TBK1 activation and oligomerization by using its kinase-inactive mutant PTK2B-K457R3. However,

it is unclear whether the kinase activity of PTK2B is essential to support IFN induction under virus infection because the authors showed both kinase activity-dependent and independent functions of PTK2B. To clarify this, the authors should examine if reconstitution of the kinase-inactive mutant (e.g., K457R) of PTK2B doesn't rescue IFN induction as shown in Fig. 3e. Also, the authors should check if the chemical inhibitor of PTK2B (PYK2/FAK2) impairs IFN and ISG expression.

10. Fig. 4f-g and 5f showed that PTK2B phosphorylates TBK1 at Y591, which promotes TBK1 oligomerization. However, it is unclear whether the phosphorylation of TBK1 at Y591 is important for antiviral responses. It should be demonstrated whether the levels of phosphorylated IRF3 and expression of antiviral genes are suppressed by reconstitution of TBK1-Y591F, but not Y591E, mutant. Alternatively, the authors could perform IFN-promoter Luciferase assays overexpressing each TBK1 mutant.

11. In Fig. 4j and 5g, the authors overexpressed the phosphomimetic mutant (Y591E) of TBK1 together with PTK2B. However, what is important here is to clarify if the phosphorylation at Y591 is sufficient to activate TBK1. Therefore, the authors should check if TBK1-Y591E induces the dimerization and activation of TBK1 without PTK2B overexpression. Also, if that is the case, they should examine if TBK1-Y591E induces IFN expression without virus infection. If not, the authors should examine if other phosphomimetic mutations would be sufficient to activate TBK1 without PTK2B overexpression or virus infection.

12. Regarding Fig. 5c, since the activation mechanism of TBK1 is different between DNA virus and RNA virus infection, the authors should check if the PTK2B-dependent dimerization of TBK1 occurs under RNA virus infection (or poly IC transfection).

Minor points

1. The authors should provide information on SDS-PAGE in the method section so that readers can understand the difference between SDD-AGE and SDS-PAGE.
2. The term "P-TBK1" in the results of WB analysis is very confusing since this study evaluates multiple phosphorylation sites of TBK1. The authors should specify the phosphorylation site in every WB data.

Reviewer #2 (Remarks to the Author):

In this manuscript, Lin et al. demonstrated that PTK2B enhances antiviral responses by regulating tyrosine phosphorylation at Tyr591 and oligomerization of TBK1 in a kinase activity-dependent manner. PTK2B depletion also affected STING oligomerization, but STING oligomerization did not require PTK2B kinase activity. Notably, PTK2B-deficient mice exhibited impaired antiviral responses, resulting in increased susceptibility to viral infection. Overall, the findings are potentially interesting. However, while the relationship between TBK1 and PTK2B has been well studied, the relationship between STING and PTK2B has not been thoroughly investigated, and the STING part of the study appears to be incomplete. Importantly, their data demonstrate that PTK2B is involved in the induction of STING oligomerization, and since STING is known to be upstream of TBK1, their data do not exclude the possibility that the effect of PTK2B on TBK1 is secondary. Therefore, the results on STING and those on TBK1 are not mutually reinforcing with the current data.

The following list of concerns should be addressed to improve the manuscript:

- The data presented in the mechanistic analysis do not clarify which step in the signaling pathway PTK2B functions. PTK2B-deficient cells exhibited impaired STING oligomerization in Figure 6C-6E, and the phosphorylation of STING tyrosine residues almost completely disappeared when using the PTK2B (K457R) mutant in Figure 6F. This observation suggests that PTK2B is indispensable for STING activation, but it is possible that the effect of PTK2B on TBK1 may be secondary. Additionally, PTK2B-deficiency affected RIG-I-mediated antiviral responses, which do not involve STING but instead involve MAVS. These findings are contradictory and not mutually reinforcing.
- In Figure 1C, while the interaction between PTK2B and TBK1 increases upon HSV-1 infection, the interaction between PTK2B and STING seems to decrease. This reviewer believes that this observation could help the authors understand the possible regulatory mechanisms of PTK2B, and they should investigate it in more detail.
- Despite the availability of KO mice, they are only used in limited experiments (cytokine qPCR, ELISA, infection experiments). Therefore, the authors should include data using cells derived from KO mice (e.g., bone marrow-derived macrophages or dendritic cells) in their mechanistic analysis, which are more physiological than current data.
- In lines 131-133, the authors describe why they chose PTK2B, but this explanation seems inadequate because IL-1beta does not activate TBK1.
- In Figure 1G, it is strange that PTK2B was not found in the sample immunoprecipitated with full-length TBK1.
- It is a bit strange that several data using PTK2B knockout mice are presented, even though the authors have access to PTK2B knockout mice.
- It would be interesting to analyze cells or mice with the Y591F mutation, as such data would support their conclusions.
- For the reader's understanding, the manuscript should include not only the product code but also a description of the antibody that specifically recognizes the phosphorylation of S172 in TBK1.
- It would be helpful for readers to include a figure at the end of the manuscript summarizing the results of the study.

REVIEWER COMMENTS

Reviewer #1 (Remarks to the Author):

In this manuscript, the authors demonstrated that protein tyrosine kinase 2 beta (PTK2B) (also known as PYK2 or FAK2) is a positive regulator of STING-TBK1 activation in antiviral innate immune responses. They showed that PTK2B interacts with TBK1 and STING, and PTK2B positively regulates expression of antiviral genes. *Ptk2b*-deficient mice were more susceptible to HSV-1 viral infection than control mice. Mechanistically, they showed that PTK2B promotes TBK1 activation and oligomerization by TBK1 phosphorylation at Y591. Additionally, PTK2B enhances STING oligomerization in a kinase-independent manner.

Although the role of PTK2B in the activation of TBK1 is potentially interesting, the importance of this protein in antiviral immunity by type I IFN stimulation looks to be limited and the analysis of the roles of PTK2B in antiviral innate immune signaling is still preliminary. Further, although the authors identified several phosphorylation sites of TBK1 by PTK2B, how each phosphorylation is important under virus infection is unclear. Specific comments are shown below.

Response: We thank the reviewer for positively commenting on our manuscript. In order to improve our work, we have performed a number of important experiments to address the reviewers' questions. The detailed responses are seen below.

Particularly, for the question about "how each phosphorylation is important under virus infection is unclear", we performed rescue experiments to examine the effects of each phosphorylation site of TBK1 by PTK2B in regulating antiviral innate immune responses upon HSV-1 infection. We found that Y591 of TBK1 is required for the full activity of TBK1 in MEF cells under virus infection. We have included the new data in the revised manuscript (Fig. S7a).

Major points

1. Although the authors show that *Ptk2b* KO mice are highly susceptible to HSV-1 infection, the increase in viral titer and the decrease of IFN β production was quite modest *in vivo* (Fig. 3). It is unlikely that such a modest impairment in IFN β production leads to the huge difference in the lethality of mice. As the authors stated, PTK2B has been reported to be involved in the activation of inflammasome (Sci Rep 2016). Thus, the authors need to clarify if the inflammasome contributes to the alteration in the lethality of HSV-1-infected *Ptk2b* KO mice.

Response: We thank the reviewer for this comment. We have repeated our experiments using *Ptk2b* KO mice infected with HSV-1 and found that the increase in viral titer and the decrease of IFN β production were indeed modest. Previous studies have suggested that both IFN β and inflammatory cytokines play antiviral roles to reduce the lethality of host mice. To address the reviewer's concern, we measured the levels of serum IL-6, and TNF α , two pro-inflammatory cytokines, from *Ptk2b*^{+/+} and *Ptk2b*^{-/-} mice after infection with HSV-1. As shown in Figs. S5a, b, the production of IL-6 and TNF α were significantly reduced in *Ptk2b* KO mice, when compared to wild type mice, upon virus

infection. Similar results were obtained when mice were infected with VSV, an RNA virus (Figs.S5g–i). Nevertheless, we can't rule out the possibility that PTK2B has a role in regulating the inflammasome signaling that potentially contributes to the alteration in the lethality of HSV-1-infected *Ptk2b* KO mice.

2. Fig. 3l showed that infiltration of immune cells and tissue damage were increased in the lungs of infected *Ptk2b*-KO mice than in WT mice. However, it is unclear what kinds of immune cells are infiltrated in lungs and whether population and activation status of infiltrated immune cells are different between *Ptk2b* WT and KO mice.

Response: To address the reviewer's question, we isolated cells from the lung of *Ptk2b*^{+/+} and *Ptk2b*^{-/-} mice after HSV-1 infection, and then performed flow cytometry. We found that while the number of macrophage was not changes between *Ptk2b*^{+/+} and *Ptk2b*^{-/-} mice, the number of DCs and Neutrophils cells was significantly increased in *Ptk2b* KO cells, when compared to that in wild type mice. We have included the new results in the revised manuscript (Figs. S5d–f)

3. It is also important to investigate the survival of *Ptk2b* KO mice in response to the infection with RNA viruses to further tighten the relationship of PTK2B with the activation of TBK1.

Response: Following the reviewer's suggestion, we infected *Ptk2b*^{+/+} and *Ptk2b*^{-/-} mice with VSV, an RNA virus, then examined the survival rates of these mice. As shown in Fig. S5i, loss of *Ptk2b* significantly reduced the survival rates of mice after VSV infection. In addition, we measured the levels of IFN β and IL-6 in serum from *Ptk2b*^{+/+} and *Ptk2b*^{-/-} mice after VSV infection, and ELISA results demonstrated that deficiency of PTK2B significantly decreased the production of these cytokines (Figs. S5g, h). We have included the new results in the revised manuscript.

4. It is also essential to uncover the mechanisms how PTK2B is activated in response to virus infection to phosphorylate TBK1 and induce STING oligomerization.

Response: Previous studies have suggested that the oligomerization of enzyme plays a critical role in regulating its activation (PMID:29976794, PMID: 30879902). Given that PTK2B and TBK1 could co-localize together and form cellular granules upon HSV-1 infection (Fig. 1d), we reasoned that PTK2B could potentially undergo an oligomerization, in response to the virus infection. To test this idea, we performed SDD-AGE assays, and observed that HSV-1 infection induced phosphorylated PTK2B oligomerization in a time-dependent manner, which is consistent with the activation of PTK2B (Fig. S9a). Moreover, we employed the "ddFP" system to study the behavior of PTK2B and TBK1 in cells. The ddFP system has two parts of monomers: one monomer (A) contains a chromophore that is quenched under the monomeric state, and the other one (B) does not form a chromophore, but it can substantially increase fluorescence of the part A, when A and B form a heterodimer. We fused PTK2B with the green fluorescence-capable (GA) domains and core (GB) domain to generate two constructs, PTK2B-GA and PTK2B-GB, respectively. As shown in Fig. 5h, in absence of viral infection, no apparent green signal was detected in cells with co-expression of PTK2B-GA and PTK2B-GB. However, strong green signals showing that PTK2B self-

interaction were detected from cellular granules, upon HSV-1 infection. In addition, TBK1 oligomerization signals were also detected to be overlapped with PTK2B granules on Golgi (Fig. 5h). TBK1 oligomerization is hallmarks for TBK1 activation. Thus, our findings uncover a mechanism by which the PTK2B oligomerization induced by viral infection play a positive role in regulating the antiviral signaling by promoting TBK1 phosphorylation and oligomerization.

5. Fig. 1 showed that PTK2B interacts with TBK1 and STING. The authors should check if PTK2B colocalizes with TBK1 granules after virus infection more clearly. Unfortunately, the scale of the images in Fig. 1d is not suitable to tell this. Additionally, it should be shown if PTK2B localizes to ER together with TBK1 and STING upon viral infection.

Response: To address the reviewer's question, we used the MEFs cells to perform additional immunostaining assays. As shown in Fig.1c, PTK2B and TBK1 clearly colocalize together and form cellular granules upon virus infection. As mentioned above, we also performed the "ddFP" assay showing that PTK2B and TBK1 (or STING) could form cellular granules together, upon HSV-1 infection (Figs. 5h, S10b). Additionally, we also observed that, upon HSV-1 infection, signals of TBK1 and STING could colocalized with the PTK2B granules on Golgi marked by GM130 in MEFs (Fig.S10c). We have included the new results in the revised manuscript.

6. The authors should also check the subcellular localization of TBK1-Y591F. Does TBK1-Y591F form granules after virus infection?

Response: To address the reviewer's question, we infected TBK1-KO MEFs with lentivirus expressing TBK1-WT and TBK1-Y591F, then infected with HSV-1, followed by immunostaining. As shown in Fig. S8b, the results showed that similar to the location of wild-type TBK1, however, the number and size of TBK1-Y591F granules induced by HSV-1 infection were markedly reduced, when compared to the wild type TBK1.

7. The authors showed that PTK2B binds to STING as well as TBK1. Since PTK2B is also involved in IFN induction under RNA virus infection (Fig. S2 and S3), they should check if PTK2B binds to MAVS or other proteins involved in RNA virus sensing.

Response: According to the reviewer's suggestion, we performed Co-IP experiments by pulling down with anti-PTK2B antibody, and found that RIG-I and MAVS failed to interact with PTK2B (Fig. S1b).

8. Fig. 2f-h and S3a-d showed that PTK2B overexpression increased the mRNA levels of antiviral genes upon infection with HSV1-GFP and VSVΔM51-GFP in MEFs. It should be shown weather the levels of phosphorylated STING, TBK1, and IRF3 are increased by PTK2B overexpression.

Response: Following the reviewer's suggestion, we have included these data in the revised manuscript (Figs. S3e, f).

9. In Fig. 4a-b and 5a-b, the authors showed that PTK2B kinase activity is required for TBK1 activation and oligomerization by using its kinase-inactive mutant PTK2B-K457R3. However, it is unclear whether the kinase activity of PTK2B is essential to support IFN induction under virus infection because the authors showed both kinase activity-dependent and independent functions of PTK2B. To clarify this, the authors should examine if reconstitution of the kinase-inactive mutant (e.g., K457R) of PTK2B doesn't rescue IFN induction as shown in Fig. 3e. Also, the authors should check if the chemical inhibitor of PTK2B (PYK2/FAK2) impairs IFN and ISG expression.

Response: We thank for the reviewer's comments and performed the rescue experiments. As shown in Fig. 3e, we found that reconstitution of the kinase-inactive mutant, PTK2B(K457R) partially rescued the decreased expression of *Irf1* induced by HSV-1-GFP in Ptk2b-deficient MEFs. These data suggested that PTK2B regulated anti-DNA viral signaling in both kinase activity-dependent and independent manner.

Regarding the issue of PTK2B inhibitor, we have tried two commercial inhibitors, including PF-562271 and NVP-TAE226, and found both of them reduced the activity of PTK2, but none of them specifically reduced the activity of PTK2B (Response Fig.1). To address the reviewer's concern, we designed an ASO (antisense-oligonucleotide) that specifically targets the human PTK2B transcripts and found that depletion of PTK2B by ASO reduced the levels of *IFNB1* mRNA induced by the infection of HSV1-GFP or VSVΔM51-GFP in A549 cells (Figs. S2h, i). Collectively, our findings strongly argue that PTK2B plays an important role in antiviral signaling.

Response Fig.1 the effects of different inhibitors on the activation of PTK2B

In response Fig. 1, Raw 264.7 cells were pretreated with the indicated inhibitor for 24 h, then infected with HSV1-GFP for the indicated time, followed by immunoblotting.

10. Fig. 4f-g and 5f showed that PTK2B phosphorylates TBK1 at Y591, which promotes TBK1 oligomerization. However, it is unclear whether the phosphorylation of TBK1 at Y591 is important for antiviral responses. It should be demonstrated whether the levels of phosphorylated IRF3 and expression of antiviral genes are suppressed by reconstitution of TBK1- Y591F, but not Y591E, mutant. Alternatively, the authors could perform IFN-promoter Luciferase assays overexpressing each TBK1 mutant.

Response: Following the reviewer's suggestion, we performed the rescue experiments in TBK1 KO MEFs. Immunoblotting assays showed that wild-type TBK1 can rescue the decreased phosphorylated IRF3 and *Irfn1* transcripts induced by HSV-1 infection, Y591E, but not Y591F, partially rescued. We have included the new data in the revised manuscript (Figs. S7c, d).

11. In Fig. 4j and 5g, the authors overexpressed the phosphomimetic mutant (Y591E) of TBK1 together with PTK2B. However, what is important here is to clarify if the phosphorylation at Y591 is sufficient to activate TBK1. Therefore, the authors should check if TBK1-Y591E induces the dimerization and activation of TBK1 without PTK2B overexpression. Also, if that is the case, they should examine if TBK1-Y591E induces IFN expression without virus infection. If not, the authors should examine if other phosphomimetic mutations would be sufficient to activate TBK1 without PTK2B overexpression or virus infection.

Response: As per the reviewer's request, we performed additional experiments, and found that in contrast to wild type TBK1, TBK1-Y591F reduced its oligomerization under the condition of PTK2B overexpression. Of note, we found that overexpression of TBK1-Y591E or TBK1-Y591D alone did not induce strong oligomerization and activation, suggesting that while the Y591 is important for the function of TBK1, the phosphomimetic mutation at Y591 (Y591E or Y591D) alone is not able to induce strong activation of TBK1 in the case of this study (Figs. 4j, 5g). Previous studies have reported that phosphorylated residue of interest is normally mutated to a negatively charged residue (e.g., aspartate or glutamate); however, these amino acids often fail to recapitulate the true steric and charge-based nature of the phosphoryl-modification (PMID 24119841, PMC 6384131)

12. Regarding Fig. 5c, since the activation mechanism of TBK1 is different between DNA virus and RNA virus infection, the authors should check if the PTK2B-dependent dimerization of TBK1 occurs under RNA virus infection (or poly IC transfection).

Response: Following the reviewer's suggestion, we performed Native-PAGE assays to detect the effect of PTK2B deficiency on the dimer of TBK1 induced by the infection of Sendai virus, (SeV, one kind of RNA virus). However, we could not detect the dimer of TBK1 with its antibody, probably due to weaker TBK1 dimerization induced by Sendai infection, compared with HSV1-GFP infection. Interestingly, we could detect the dimer of phosphorylated TBK1, probably due to higher sensitivity of phosphorylated TBK1 antibody than TBK1 antibody in our hands. As shown in Fig. S8a, we observed that Ptk2b deficiency reduced the dimer formation of phosphorylated TBK1 induced by the infection of Sendai virus in Raw264.7 cells. These findings further support that PTKB positively regulates TBK1 dimerization induced by virus infection.

Minor points

1. The authors should provide information on SDS-PAGE in the method section so that readers can understand the difference between SDD-AGE and SDS-PAGE.

Response: We have included the detailed information in the method section.

2. The term “P-TBK1” in the results of WB analysis is very confusing since this study evaluates multiple phosphorylation sites of TBK1. The authors should specify the phosphorylation site in every WB data.

Response: We have included the required information in the revised manuscript.

Reviewer #2 (Remarks to the Author):

In this manuscript, Lin et al. demonstrated that PTK2B enhances antiviral responses by regulating tyrosine phosphorylation at Tyr591 and oligomerization of TBK1 in a kinase activity-dependent manner. PTK2B depletion also affected STING oligomerization, but STING oligomerization did not require PTK2B kinase activity. Notably, PTK2B-deficient mice exhibited impaired antiviral responses, resulting in increased susceptibility to viral infection. Overall, the findings are potentially interesting. However, while the relationship between TBK1 and PTK2B has been well studied, the relationship between STING and PTK2B has not been thoroughly investigated, and the STING part of the study appears to be incomplete. Importantly, their data demonstrate that PTK2B is involved in the induction of STING oligomerization, and since STING is known to be upstream of TBK1, their data do not exclude the possibility that the effect of PTK2B on TBK1 is secondary. Therefore, the results on STING and those on TBK1 are not mutually reinforcing with the current data.

Response: We thank the reviewer for positively commenting on our manuscript. Regarding the issue of the relationship between PTK2B and STING, in the original manuscript, we have performed several experiments to investigate their regulatory relationship. 1) Co-IP experiments showed that PTK2B directly interacted with STING, 2) SDD-AGE and immunostaining showed that PTK2B enhanced STING oligomerization in a kinase-independent manner. In the revision, we have performed additional experiments to further analyze their interaction at the subcellular levels. By conducting immunostaining assays, we showed that PTK2B and STING could co-localize together and form cellular granules on Golgi upon HSV-1 infection (Fig.S10b). Importantly, knockout of PTK2B significantly reduced the size and number of STING granules induced by HSV-1 infection (Fig. 6d). Collectively, these data suggest that PTK2B associated with STING, subsequently enhanced STING oligomerization in a kinase-independent manner. We have included the new results in the revised manuscript.

Regarding the question about whether the effect of PTK2B on TBK1 is secondary, we have provided several lines of evidence showing that PTK2B can directly regulate TBK1 oligomerization and activation. **First**, we used HEK293T cells to perform Co-IP experiments, and found that PTK2B co-immunoprecipitated with TBK1 (Fig.1b). Because STING did not express in HEK293T cells, our results suggest that PTK2B associates with TBK1 in a STING-independent manner. **Second**, we used the purified PTK2B and TBK1 proteins from *E. Coli.*, and performed in vitro pull down experiments.

We again found that PTK2B directly interacts with TBK1 (Fig. 1e). **Third**, we co-expressed TBK1 and PTK2B in HEK29T cells, performed SDD-AGE assays, and found that PTK2B can increase TBK1 oligomerization (Fig. 5a). Consistent results were obtained when we used purified PTK2B and TBK1 protein from *E.Coli*. in the SDD-AGE assays (Fig. 5b). **Forth**, in the kinase assays, our in vivo and in vitro assays showed that that PTK2B can phosphorylate TBK1 in the absence of STING (Figs. 4b and 4c), suggesting that PTK2B can directly regulate TBK1 oligomerization and phosphorylation in a STING-independent manner.

The following list of concerns should be addressed to improve the manuscript:

- The data presented in the mechanistic analysis do not clarify which step in the signaling pathway PTK2B functions. PTK2B-deficient cells exhibited impaired STING oligomerization in Figure 6C-6E, and the phosphorylation of STING tyrosine residues almost completely disappeared when using the PTK2B (K457R) mutant in Figure 6F. This observation suggests that PTK2B is indispensable for STING activation, but it is possible that the effect of PTK2B on TBK1 may be secondary. Additionally, PTK2B-deficiency affected RIG-I-mediated antiviral responses, which do not involve STING but instead involve MAVS. These findings are contradictory and not mutually reinforcing.

Response:

For the issue of whether the effect of PTK2B on TBK1 is secondary, we have answered the question above.

Regarding the issue about whether PTK2B regulates RIG-I-mediated antiviral responses through affecting MAVS, we performed Co-IP experiments and SDD-AGE assays, and found that PTK2B did not associate with MAVS (Fig. S1b), its overexpression did not affect MAVS oligomerization (Response Fig. 2). Given that TBK1 is an important regulator in the RIG-I-MAVS-mediated antiviral signaling, and that PTK2B enhances TBK1 activation, we argue that PTK2B modulates the RIG-I-MAVS-mediated antiviral signaling mainly through regulating TBK1.

Response Fig. 2 PTK2B did not enhance MAVS oligomerization

In response Fig. 2, HEK293T cells were co-transfected with the indicated plasmids. Cell lysates were resolved by semi-denaturing detergent agarose gel electrophoresis (SDD-AGE) and sodium dodecyl sulfate-polyacrylamide gel electrophoresis (SDS-PAGE), followed by immunoblotting.

- In Figure 1C, while the interaction between PTK2B and TBK1 increases upon HSV-1 infection, the interaction between PTK2B and STING seems to decrease. This reviewer believes that this observation could help the authors understand the possible regulatory mechanisms of PTK2B, and they should investigate it in more detail.

Response: Previous studies demonstrated that HSV-1 virus infection can induced STING degradation, and MG132 (a proteasome inhibitor) can block STING degradation induced by HSV-1 infection (PMID: 27428826). To avoid the interference of STING degradation induced by HSV-1, we pretreated cells with MG132 to prevent STING degradation. As shown by Co-IP analysis, HSV-1 infection had no obvious effect on the interaction of PTK2B-STING. We have included the new results in the revised manuscript (Fig. S1a).

- Despite the availability of KO mice, they are only used in limited experiments (cytokine qPCR, ELISA, infection experiments). Therefore, the authors should include data using cells derived from KO mice (e.g., bone marrow-derived macrophages or dendritic cells) in their mechanistic analysis, which are more physiological than current data.

Response: In the original version of manuscript, except MEFs, we have also included the results by employing bone marrow-derived macrophages (BMDM) (Fig. 3f-h and Fig. S4d-f). In the revision, following the reviewer's suggestions, we also conducted the additional experiments using dendritic cells from *Ptk2b*^{+/+} and *Ptk2b*^{-/-} mice, obtained the consistent results with MEFs and BMDM cells. We have included the new results in the revised manuscript (Figs. S4g-l).

- In lines 131-133, the authors describe why they chose PTK2B, but this explanation seems inadequate because IL-1beta does not activate TBK1.

Response: We want to point out that PTK2B was identified by the co-IP with TBK1 followed by mass spec assays, suggesting that PTK2B is a TBK1-associated protein. Moreover, phosphorylation of TBK1 is important for its kinase activity and thus for transducing the TBK1 signaling. Given that PTK2B is a kinase, we were interested to know whether this kinase is involved in the regulation of TBK1. We have made a change in the revised manuscript.

- In Figure 1G, it is strange that PTK2B was not found in the sample immunoprecipitated with full-length TBK1.

Response: When we co-transfected PTK2B and TBK1, we noticed that TBK1 can phosphorylate PTK2B, resulting in a shift of PTK2B in the mobility (Figs.1a, b). In Fig.1g, PTK2B displayed a shift band when it was co-expression with TBK1 in IP samples.

- It is a bit strange that several data using PTK2B knockout mice are presented, even though the authors have access to PTK2B knockout mice.

Response: In the original version of manuscript, we employed PTK2B knockout mice and performed a series of experiments. First, we isolated MEFs from *Ptk2b*^{+/+} and *Ptk2b*^{-/-} mice, and performed q-PCR and immunoblotting assays, and found that PTK2B deficiency reduced antiviral signaling upon HSV-1 and VSV infection. Second, we obtained consistent results when we examined BMDM cells from *Ptk2b*^{+/+} and *Ptk2b*^{-/-} mice. Third, we infected *Ptk2b*^{+/+} and *Ptk2b*^{-/-} mice with HSV-1 and measured the production of IFN β , IL-6 and TNF α , virus replication, and survival rates of infected mice and also perform HE staining to examine the damage of lung. These results provided *in vivo* evidence that PTK2B was required for efficient defense against HSV-1 infection. In the revised manuscript, we also tested the effects of PTK2B depletion on the antiviral signaling in dendritic cell isolated from *Ptk2b*^{+/+} and *Ptk2b*^{-/-} mice and obtained the consistent results with MEFs and BMDM cells (Figs. S4g-l). Additionally, we infected *Ptk2b*^{+/+} and *Ptk2b*^{-/-} mice with VSV, one kind of RNA virus, and demonstrated that PTK2B knockout reduced the production of IFN β , IL-6 and the survival rates of mice (Figs. S5g-i). Collectively, these findings suggest that PTK2B plays a critical role in maintaining efficient defense against virus infection.

- It would be interesting to analyze cells or mice with the Y591F mutation, as such data would support their conclusions.

Response: Following the reviewer's suggestion, we performed the rescue experiments by using TBK1 KO MEFs. As shown in Figs.S7a, c showed that unlike TBK1-WT, TBK1-Y591F, only partially rescued the decreased phosphorylated IRF3 and *Ifnb1* mRNA induced by HSV-1 infection. Consistently, immunostaining results showed that the number and size of TBK1-Y591F granules induced by HSV-1 infection were markedly reduced, when compared to the wild type TBK1 (Fig. S8b). We have included the new data in the revised manuscript. These data suggest that Y591 of TBK1 is

phosphorylated by PTK2B and this phosphorylation plays an important role in regulating the full activation of TBK1.

- For the reader's understanding, the manuscript should include not only the product code but also a description of the antibody that specifically recognizes the phosphorylation of S172 in TBK1.

Response: We have included the required information in the revised manuscript.

- It would be helpful for readers to include a figure at the end of the manuscript summarizing the results of the study.

Response: We have included the required information in the revised manuscript.

REVIEWERS' COMMENTS

Reviewer #1 (Remarks to the Author):

The revised manuscript is substantially improved. I think now this manuscript is ready for publication.

Reviewer #2 (Remarks to the Author):

I have carefully reviewed the revised manuscript, and would like to commend the authors for their diligent efforts in addressing the concerns raised during the initial review process. The revisions made have significantly improved the quality and clarity of the manuscript. Furthermore, the additional experiments and data provided in response to my comments have strengthened the validity and robustness of the research findings. I am pleased to inform that, based on the substantial improvements made in the revised manuscript, I recommend its acceptance for publication.

Reviewer #1 (Remarks to the Author):

The revised manuscript is substantially improved. I think now this manuscript is ready for publication.

Response: We thank the reviewer for supports.

Reviewer #2 (Remarks to the Author):

I have carefully reviewed the revised manuscript, and would like to commend the authors for their diligent efforts in addressing the concerns raised during the initial review process. The revisions made have significantly improved the quality and clarity of the manuscript. Furthermore, the additional experiments and data provided in response to my comments have strengthened the validity and robustness of the research findings. I am pleased to inform that, based on the substantial improvements made in the revised manuscript, I recommend its acceptance for publication.

Response: We thank the reviewer for supports.